# Correlative molecular-to-mesoscale evolution in conjugated polymers for intrinsically stretchable organic photovoltaics

Wenkai Zhong [1,2] ✉, Guillaume Freychet [3,4], Gregory M. Su [5,6], Siyi Wang[1], Xuanang Luo[1], Xinrui Liu[1], Wenyu Yang[1], Lei Yu[1], Xuefei Wu[6], Yulong Li[1], Thomas J. Ferron [5], Thomas P. Russell [6,7], Lei Ying [1], Fei Huang [1], Yongming Zhang[2], Cheng Wang [5] ✉ & Feng Liu [2] ✉

Conjugated polymer thin films offer a unique combination of tunable optoelectronic properties and mechanical flexibility, making them as promising materials for intrinsically stretchable optoelectronic devices. However, achieving both mechanical robustness and high device performance remains a key challenge. Addressing this requires a fundamental understanding of how molecular and mesoscale structures evolve under mechanical strain. Here, we employ a comprehensive suite of X-ray spectroscopy and scattering techniques to investigate the multiscale structural evolution of conjugated polymer thin films during uniaxial deformation. We uncover a two-stage morphological response: an initial stage characterized by polymer chain alignment and rapid crystallite disruption, followed by continued chain orientation accompanied by intrachain torsion at higher strains. These correlative structural adaptations govern key material properties, including stress dissipation, optical absorption, and photovoltaic performance. Our findings establish a mechanistic framework for understanding deformation in semiconducting polymers and provide design principles for developing mechanically robust, high-performance stretchable electronics.

Intrinsically stretchable organic semiconductor electronics have shown significant interest for applications in wearable sensors, volumetric displays, and energy-harvesting devices[1–9]. Thin films of π-conjugated polymers are of particular interest in these systems, because their extended π-π stacking and chain statistical flexibility endow them with good optoelectronic performance even under applied strain. However, understanding how deformation drives polymeric structural evolution at various scales remains challenging, since the semi-flexible backbones and high torsional energy barriers of π-conjugated polymers result in mechanical properties far inferior to

[1]Institute of Polymer Optoelectronic Materials and Devices, Guangdong Basic Research Center of Excellence for Energy and Information Polymer Materials, State Key Laboratory of Luminescent Materials and Devices, South China University of Technology, Guangzhou, China. [2]Frontiers Science Center for Transformative Molecules, Center of Hydrogen Science, School of Chemistry and Chemical Engineering, Shanghai Jiao Tong University, Shanghai, China. [3]NSLS-II, Brookhaven National Laboratory, Upton, NY, USA. [4]Univ. Grenoble Alpes, CEA, Leti, F-, Grenoble, France. [5]Advanced Light Source, Lawrence Berkeley National Laboratory, Berkeley, CA, USA. [6]Materials Sciences Division, Lawrence Berkeley National Laboratory, Berkeley, CA, USA. [7]Polymer Science and Engineering Department, University of Massachusetts, Amherst, MA, USA. ✉e-mail: wkzhong@scut.edu.cn; cwang2@lbl.gov; fengliu82@sjtu.edu.cn

those of flexible polymers[10–14]. It is therefore critical to elucidate the correlations among these multiscale structural changes to establish robust structure-property relationships for emerging stretchable devices.

Prior studies suggested that applied stress in deformed conjugated polymer thin films is dissipated via crystallite breakage, chain orientation, and even disruption of chemical bonds[15–17]. For conventional flexible polymers under deformation, strain is relieved by relaxation across multiple length and time scales, a behavior well described by Gaussian-chain models. The rigid backbones and high torsional barriers of π-conjugated polymers, however, severely restrict in-chain conformation coordination. Thus, elastic deformation is minimal, and the large conformational changes are typically accompanied by crystallographic plane dissociation once the yield point is exceeded[18]. Nevertheless, conjugated polymers have enabled intrinsically stretchable electronics by molecular design and microstructural control. In organic field-effect transistors (OFETs), high-molecular-weight polymers shows retained crystalline domains and reversible chain alignment under >100% strain, thereby preserving electrical performance[19]. Random terpolymer design can suppress crystallinity and long-range order, while maintaining short-range ordered aggregates, resulting in high mobility with improved stretchability[20]. In organic photovoltaics (OPVs), polymer-polymer blends outperform polymer-small molecule blends because gradual chain alignment and entangled networks dissipate stress while sustaining charge transport pathways[21]. The addition of high-molecular-weight conjugated polymers offers soft, amorphous regions with entangled and tie chains for stress dissipation, shielding the brittle acceptor crystallites and mitigating efficiency loss[17]. In organic light-emitting diodes (OLEDs), plasticizer incorporation transforms brittle conjugated polymers into fibrillar morphologies, resolving the conductivity-stretchability trade-off[22]. Despite these advances, direct in situ probing of the multiscale structural changes in conjugated polymer thin films, and how they collectively govern stress dissipation from the molecular to the mesoscale, remains an outstanding challenge.

Recent advancements in stretchable polymer semiconductors and energy-tunable X-ray characterization tools now offer a path to address this challenge directly[23–34]. The current work combines high-brightness soft and tender X-ray techniques to unravel the deformation behavior of conjugated polymer thin films from molecular to the mesoscopic scale. Near-edge X-ray absorption fine structure (NEXAFS) spectroscopy reveals strain-induced backbone orientation and intrachain torsion; tender resonant X-ray scattering (TReXS) demonstrates crystallite deformation; and resonant soft X-ray scattering (RSoXS) tracks morphological evolution at larger length scales. We identify a two-step morphological response that modulates mechanical behaviors, which influence absorption and photovoltaic performance. These insights into correlative, multiscale structural adaptation provide a foundation for rational design of durable and high-performance stretchable electronics.

## Results

### Molecular orientation and torsion
We selected the donor-acceptor copolymer poly[(N,N′-bis(2-octyldodecyl)-naphthalene-1,4,5,8-bis(dicarboximide)-2,6-diyl)-alt-5,5′-(2,2′-bithiophene)] (P(NDI2OD-T2)) as a model polymer for exploring the structural changes of stretchable polymer thin films (Fig. 1a). This polymer comprises alternating bithiophene (BT) and naphthalene diimide (NDI) units, with a dihedral angle of ~45° between NDI and BT (Supplementary Fig. 1). The ~200 nm film shows stretchable without cracks up to 50% strain when supported on a polydimethylsiloxane (PDMS) membrane (Supplementary Fig. 2).

To study the structures at the molecular level, NEXAFS spectroscopy was performed with a film-on-elastomer sample (Supplementary Fig. 3a). The emitted photons from the stretching thin film were collected by a 2D detector, resulting in total fluorescence yield (TFY) spectra with bulk-sensitivity (probing depth: ~100 nm) (Supplementary Fig. 4, 5). The NEXAFS intensity depends on the angle γ between the electrical field $\vec{E}$ of incident X-ray and the electronic transition dipole moment (TDM, $\vec{O}$) within the chemical bonds: $I \propto |\vec{E} \cdot \vec{O}|^2 \propto (\cos\gamma)^2$ (Fig. 1b)[35]. As shown in Fig. 1c, the TFY of polymer shows multiple peaks at 283.6–286.0 eV, corresponding to C 1s→π* transitions with TDMs normal to the conjugated planes. Spectral simulations reveal that the peaks at 283.6–284.4 eV mainly come from transitions localized to the NDIs, while the peaks at 284.4–286.0 eV arise from transitions delocalized across the NDI and BT units (Supplementary Figs. 6, 7)[36].

We define the stretch direction (SD) of the film as the $x$ axis, the transverse direction within the film plane as the $y$ axis, and the surface normal as the $z$ axis. Linearly polarized X-rays were incident at a fixed angle $θ$, with two distinct polarizations: $\vec{E}$ parallel to the $x$ axis (horizontal polarization, H.) and $\vec{E}$ perpendicular to the $x$ axis (vertical polarization, V.). These polarization conditions enable directional NEXAFS measurements to probe chain orientations along the in-plane (IP) and out-of-plane (OOP) directions. Under our experimental geometry, the NEXAFS spectra along the x-, y-, and z-axes were approximated by measurements using the H. beam, the V. beam at an incident angle of 80°, and the V. beam at an incident angle of 20°, respectively. The x-, y-, and z-spectra were used to calculate the refractive indices[37], including both the dispersive (δ) and absorptive (β) components (δ is shown in Supplementary Fig. 8). In the pre-stretched film (strain = 0%), the β values at 283.6–286.0 eV show a trend with $β_z > β_x ≈ β_y$ (Fig. 1d). In contrast, for the 50% strain film, the β values at 283.6–286.0 eV exhibit a trend with $β_z > β_y > β_x$ (Fig. 1e). To interpret these results, we simulated the π* transition intensities by aligning the polymer backbone along the $x$ axis, which yield a trend of $I_z > I_y > I_x$ within the 283.6–286.0 eV range (Supplementary Fig. 9). The comparison between experiment and simulation confirms that, upon stretching, chain backbones orient in the SD in the IP direction while remain face-on in the OOP direction.

To assess OOP orientation changes, angle-dependent NEXAFS spectra at incident angles ($θ$) of 20°, 55°, and 80° were measured (Fig. 1f). At 0% strain, the tilt angle ($α$) of π* TDMs relative to the surface normal was extracted using the relationship of $I = \frac{A}{3}[1 + \frac{1}{2}(3\cos^2\theta - 1)(3\cos^2\alpha - 1)]$, where $I$ is the peak area, and A is a constant[35,38]. This angular dependence of NEXAFS intensity is appropriate for samples with three-fold or higher substrate symmetry, consistent with the as-cast P(NDI2OD-T2) thin films prepared by spin-coating. While some conjugated polymer systems can indeed exhibit IP alignment depending on the film-forming process[38], we note that this fitting method should not be considered universally valid for all thin films prior to stretching. Under 50% strain, the film became a two-fold rotational symmetry in the IP direction. The $α$ angle can be fitted from the expression $I = A(\cos^2\theta\cos^2\alpha + \sin^2\theta\sin^2\alpha\cos^2\varphi)$, where the $φ$ is the azimuthal angle of π* TDMs in the IP direction. The calculated $α$ angles at Area 1 (π* signals localized in NDI, 283.6–284.4 eV) and Area 2 (total π* signals, 283.6–286.0 eV) are 32.2° and 34.6°, respectively, which are 5.6° and 8.0° decrease compared to the pre-stretched film (Fig. 1g, and Supplementary Fig. 10–13). The larger decrease in intensity for Area 2 compared to Area 1 suggests that the NDI and the entire polymer chain showed different motions in response to strain. Specifically, the BT shows greater intrachain rotation relative to the substrate than the NDI, suggesting a higher torsion angle between BT and NDI.

The intrachain torsion development as a function of strain was evaluated by NEXFAS spectra measured using horizontal X-rays during incremental film strains (Supplementary Fig. 14). With increasing strain, the intensity ratio of Area 1 relative to Area 2 increases and becomes larger at high strains (Fig. 1h). Molecular dynamics (MD) simulations corroborate such observation, showing an exponential rise in the dihedral angle between BT and NDI with increasing strain

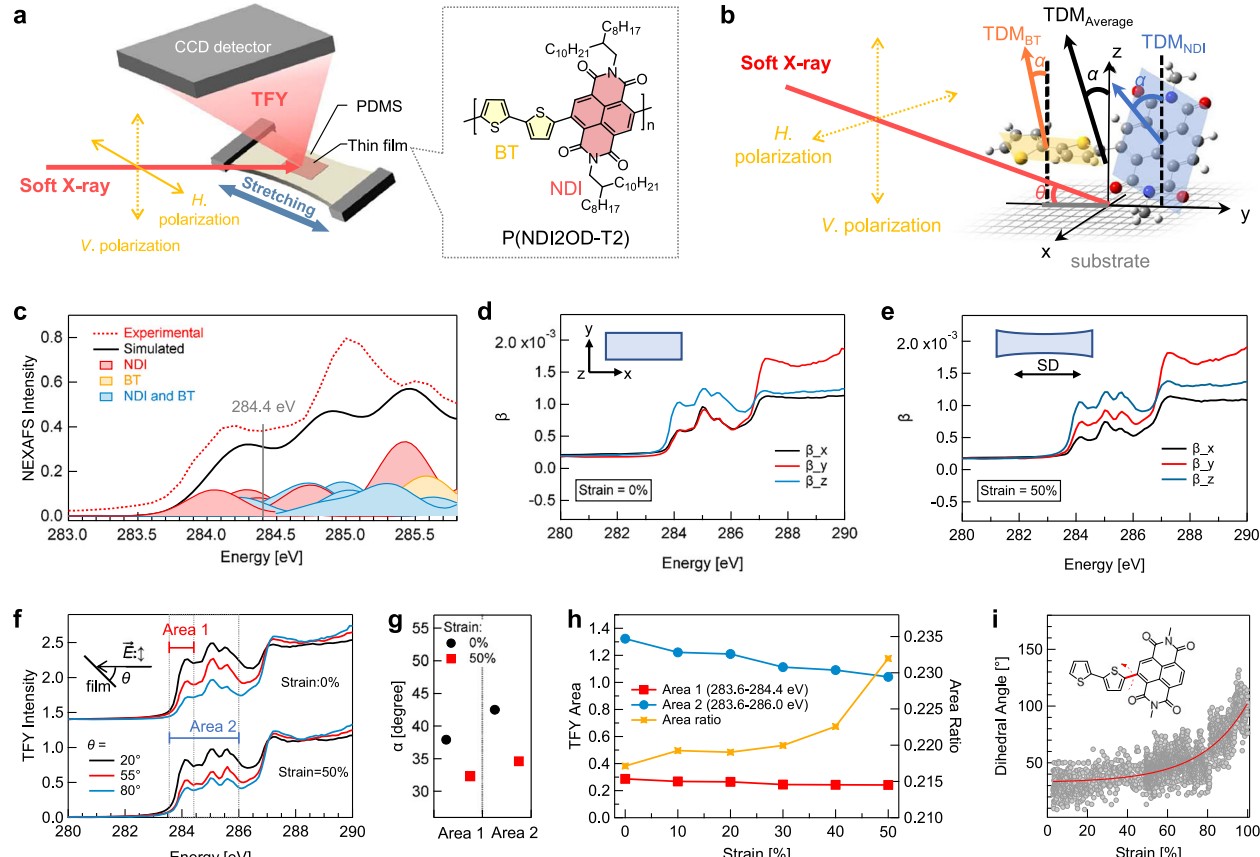

**Fig. 1 | Chain orientation and intrachain torsion. a** Experimental setup for NEXAFS spectroscopy using a film-on-elastomer sample. **b** Schematic illustration of incident soft X-rays with horizontal (*H*.) or vertical (*V*.) polarization, and the TDMs of the conjugated planes in P(NDI2OD-T2). (**c**) Experimental (TFY) and simulated NEXAFS spectra of P(NDI2OD-T2) in the C 1s→π* transitions range (283.6–286.0 eV), and the contributions from the NDI and BT units. Directional refractive indices (β) for P(NDI2OD-T2) thin films under **d** pre-stretched and (**e**) 50% strain conditions. **f** Angle-dependent NEXAFS spectra of P(NDI2OD-T2) thin films in both pre-stretched and 50% strain conditions. **g** Fitted tilt angles (α) for Area 1 (283.6–284.4 eV) and Area 2 (283.6–286.0 eV). **h** The intensity developments of Area 1, Area 2, and the Area 1/Area 2 ratio in NEXAFS spectra of the P(NDI2OD-T2) thin film under increasing strains. **i** Changes in the dihedral angle between the NDI and BI units during P(NDI2OD-T2) chain stretching, as calculated by MD simulations.

(Fig. 1i, and Supplementary Fig. 15). Achieving high dihedral angles requires surmounting specific energy barriers between subunits as demonstrated by potential energy surface scans (Supplementary Fig. 16). Together, these results indicate that strain-induced chain orientation is accompanied with intrachain conformational torsion, which plays a critical role in dissipating stress under high strains.

## Crystallite orientation, deformation and destruction

To study crystallite orientation and deformation, we employed TReXS in a transmission geometry with horizontal X-rays at 2476 eV (Fig. 2a). The PDMS-supported freestanding configuration provides a practical approach to impose uniaxial strain on thin polymer films during in situ X-ray scattering experiments, as validated by simulated strain and stress maps (Supplementary Figs. 17, 18). The selected photon energy of 2476 eV enables the selective enhancement of the (001) reflection along the SD due to the S 1s→σ* transition localized to the thiophenes (Supplementary Figs. 19–21)[39,40]. P(NDI2OD-T2) has two main reflections: a (100) reflection at q~0.25 Å⁻¹ and a (001) reflection at q~0.47 Å⁻¹, corresponding to lamellar stacking and in-chain electronic interaction (Fig. 2b)[41]. As shown in Fig. 2c, we categorize crystallites as either parallel or perpendicular based on the orientation of their chain backbones relative to the SD. Parallel crystallites have chain backbones aligned with the SD, displaying (001) and (100) reflections in the $q_x$ (∥) and $q_y$ (⊥) directions, respectively. Perpendicular crystallites with backbones transverse to the SD, show (001) and (100) reflections in the $q_y$ and $q_x$ directions, respectively.

Under 0–33% increasing strain, the parallel crystallites exhibit a 37% drop in (001) intensity and a 44% drop in (100) intensity (Fig. 2d, and Supplementary Fig. 22–25). In contrast, the perpendicular crystallites show more significant degradation, with (001) intensity decreasing by 58% and (100) by 54%. Such observations are particularly pronounced in the early stage of stretching. Upon releasing the film, the scattering intensities continue to decrease between 30% and 20% strain, likely due to residual stress remaining above the critical threshold for crystallite destruction. Below ~20% strain, no further decrease is observed; however, neither reflection recovers, indicating a permanent loss of crystallinity for both directions. The (001) anisotropic ratio, defined as $(I_∥ - I_⊥)/(I_∥ + I_⊥)$[42], increases from 0.12 to 0.33 during stretch (Supplementary Fig. 26). These results suggest that crystallites are highly strain-sensitive, with parallel crystallites being less susceptible to strain-induced destruction, resulting in enhanced crystallite orientation in the SD.

To further explore crystallite deformation, we calculated the layer packing number for each reflection by taking the ratio of crystalline coherence length (CCL) to d-spacing (Fig. 2e). For parallel crystallites, the packing number for the (100) reflection decreases significantly, while the (001) reflection remained stable, suggesting crystallographic slippage, where the layers slide past one another along the SD

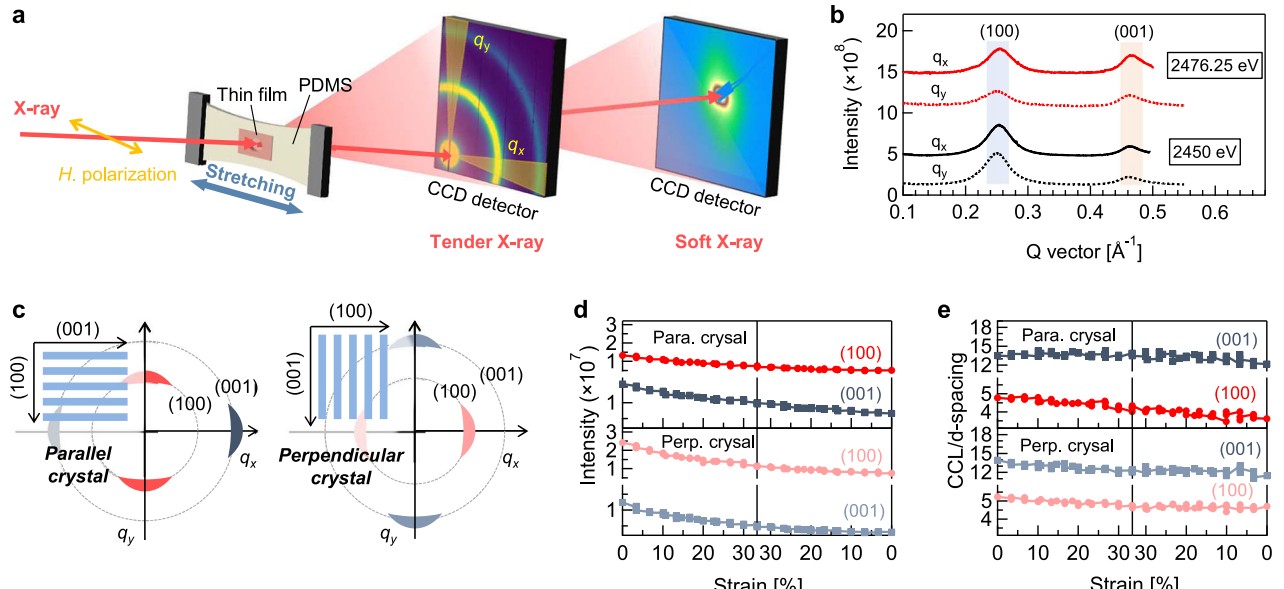

**Fig. 2 | Crystallite deformation and destruction. a** Experimental setup for TReXS and RSoXS in transmission geometry, where TReXS probes crystalline packing and RSoXS reveals the morphology of the P(NDI2OD-T2) thin films. **b** TReXS *I-q* curves averaged along the $q_x$ and $q_y$ directions at 2450 eV and 2476.25 eV. **c** Schematic illustrations of (100) and (001) reflections for parallel and perpendicular crystallites. **d** Intensity and (**e**) packing number (CCL/*d*-spacing) of the (100) and (001) reflections as functions of strain for parallel (para.) and perpendicular (perp.) crystals.

(Supplementary Fig. 27a). Such slippage allows the crystallite layers to partially shift without entirely losing their stacked arrangement, thus accommodating strain while keeping some structural integrity. In contrast, perpendicular crystallites showed that packing number for the (001) reflection decreased more noticeably, suggesting that more repeated units are lost from the crystallites, a process we term crystallographic peeling (Supplementary Fig. 27b). Rather than sliding, the crystallite layers pull apart at the edges, resulting in a greater and faster structural disruption, while providing additional amorphous chains that align and undergo intrachain torsion.

To study the π-π stacking developments during stretch, the films were stretched to the desired strains and then placed on silicon wafers for grazing incidence wide angle X-ray scattering (GIWAXS) measurements at the beam energy of 10 keV. This measurement was performed with orthogonal X-ray directions including parallel and perpendicular conditions (Supplementary Fig. 28). The 0% strain thin film shows nearly identical reflections (Supplementary Fig. 29), which support that $\beta_x \approx \beta_y$ before stretching as demonstrated by NEXAFS. Meanwhile, the (100) peak area in both $q_x$ and $q_y$ decreases rapidly at small strains (10-20%) and then more gradually at higher strains, with a stronger reduction along $q_x$ (Supplementary Fig. 30–32, and Supplementary Table 1-2). This behavior is consistent with the observations at 2476.25 eV in TReXS, indicating that the resonant excitation is not sensitive to the crystalline deformation. The azimuthal intensity of the π-π stacking reflection was used to evaluate crystallite orientation relative to the substrate (Supplementary Fig. 33). With increasing strain, the pole figures show narrowing for the perpendicular condition and broadening for the parallel condition. Therefore, an enhanced face-on orientation of π-π stacking is observed when X-rays probe perpendicular to the SD, in agreement with the angle-resolved NEXAFS results. The opposite trend in pole figure widths suggests reduced twisting of crystallites about the (001) axis but increased tilting about the (100) axis, consistent with the tilted crystallite orientation observed in flow-induced alignment[42]. Together, these combined mechanisms of crystallite deformation and orientation provide irreversible structural adaptation pathways for stress dissipation.

## Morphological evolution under strain

The chain orientation with intrachain torsion and crystallite destruction set the basis for large-scale morphological changes in the film, which we studied using in situ RSoXS in a transmission geometry (Fig. 2a). To optimize the measurements, we estimated the scattering contrast to determine an appropriate soft X-ray energy. As shown in Fig. 3a, atomic force microscopy (AFM) height images reveal the pre-stretched film showing refined fibrillar nanostructures. For the 50% strained film, large-scale fibril aggregates with their long axes align toward the SD are seen, showing a fast Fourier transform (FFT) pattern with an X-shaped anisotropy, which results from the progressive development in both size and orientation of the aggregates (Supplementary Fig. 34). This reflects the formation of a statistically oriented structure induced by strain. We then calculated the orientational contrast in the xy, xz, and yz planes by $E^4(\Delta\delta^2 + \Delta\beta^2)$ based on refractive indices δ and β along each axis (Fig. 3b)[43,44]. This analysis shows a significant intensity increase for the xy- and xz-components at ~284.0 eV, suggesting that chain backbone orientations in both IP and OOP directions strongly influence the scattering signal. Further analysis of the integrated scattering intensity (ISI, ISI $\propto \int_{q_1}^{q_2} Iq^2 \, dq$) from the RSoXS *I-q* curves confirms that at ~284.0 eV, scattering contrast is maximized when the free-standing film is strained to 30% (Fig. 3c, and Supplementary Fig. 35–38).

RSoXS patterns probed at 284.0 eV during a stretch-release process (stretch: 0-30%; release: 30-0%) are shown in Fig. 3d, e. At the pre-stretched state (0%), the film shows a slightly anisotropic scattering pattern that varies with X-ray polarization (Supplementary Fig. 39), suggesting contributions from local molecular orientation or the form factor of core-shell-like fibrillar domains[32,45]. Under mild strain (5.66%), the scattering pattern changed to an X shape, which elongates transversely with increasing strain (5.66-29.81%). This X-shaped pattern is independent of X-ray polarization (Supplementary Fig. 40), indicating that global orientation occurs upon stretching. The corresponding characteristic domain size is also evaluated using the Debye-Bueche model, which shows a trend of structural reorganization that complements the micro-scale

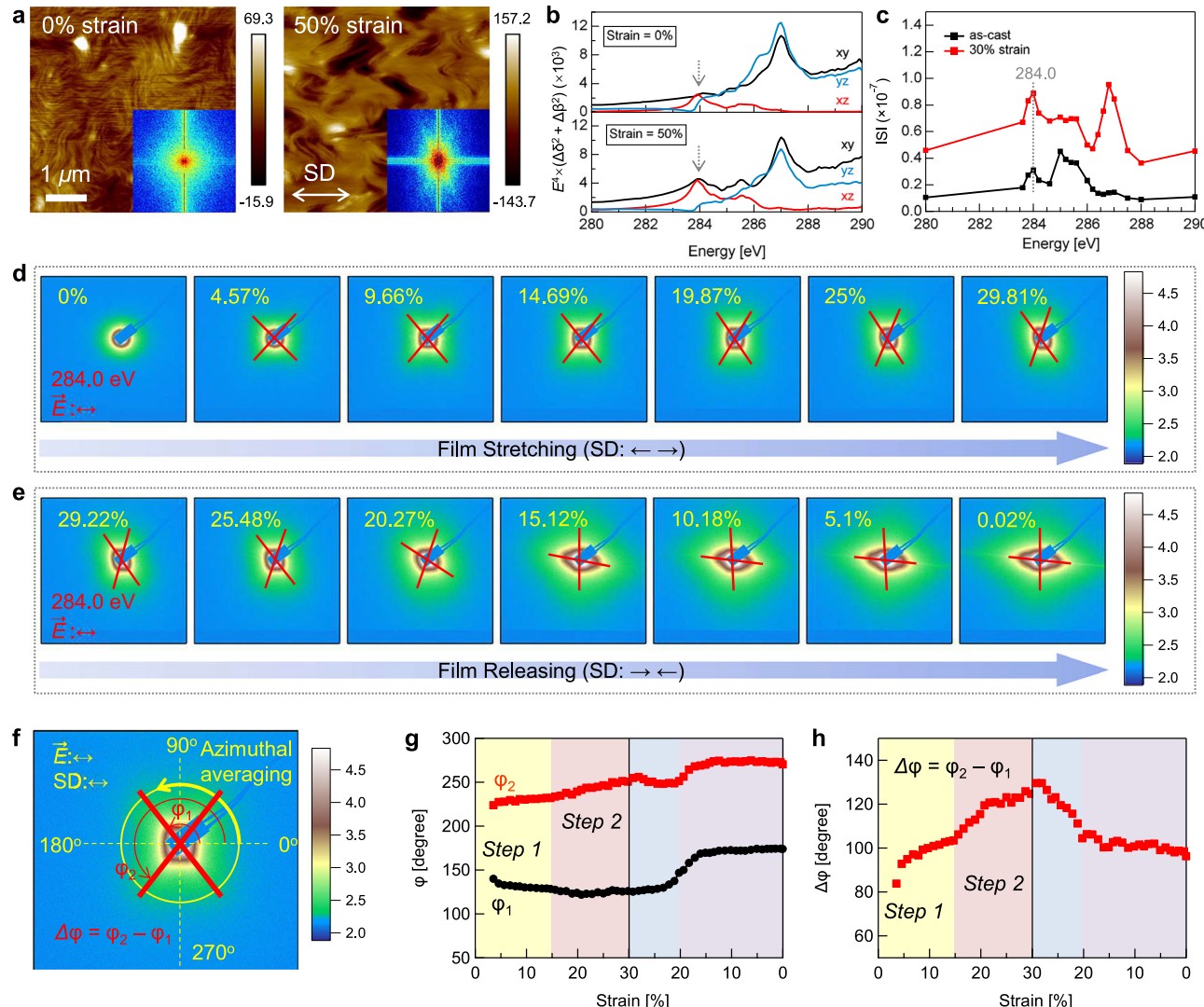

**Fig. 3 | Morphological response under tensile deformation. a** AFM height images of P(NDI2OD-T2) thin films under pre-stretched (0% strain) and 50% strain conditions. Insets are the FFT images of the corresponding AFM images. The color bar represents the vertical height variation of the sample surface, with values in nanometer. **b** Orientational contrast in the xy, xz, and yz planes for P(NDI2OD-T2) thin films under pre-stretched (0% strain) and 50% strain conditions. **c** Energy-dependent ISIs extracted from RSoXS of the P(NDI2OD-T2) thin film under 0% and 30% strain. The ISIs were calculated by integrating the $Iq^2$-$q$ curves over the $q$ range of 0.001-0.01 Å$^{-1}$. Data with $q$ greater than 0.01 Å$^{-1}$ was excluded for minimizing fluorescence background that could distort the peak features of the ISI curves. RSoXS images of a P(NDI2OD-T2) thin film at different strains during (**d**) stretching and (**e**) releasing. The color bar reflects the variation in scattering intensity. **f** Illustrations of the angular positions ($\varphi_1$ and $\varphi_2$) of the X-shaped scattering signals. The developments of (**g**) $\varphi$ and (**h**) $\Delta\varphi$ as functions of strain during a stretching-releasing process.

evolution observed by AFM (Supplementary Figs. 41–43). We note that the strained thin-film morphology is complex, as domain and orientational contrasts are intertwined and difficult to decouple. Nevertheless, the consistent X shape observed in AFM FFT images, the enhanced orientational contrast in the xy- and xz-planes at 284.0 eV, and the local maximum ISI at 284.0 eV indicate that the X-shaped scattering arises from strain-induced alignment of polymer chains across the fibrillar morphology. In situ RSoXS measurements at 285.0 eV during stretching show that X-shaped feature disappears (Supplementary Fig. 44), suggesting reduced orientation sensitivity and further supporting the assignment of scattering contrast origin. Thus, the X pattern provides direct information related to the orientation of fibril aggregates. Such results also indicates that although the crystalline scattering exhibits similar strain decay profiles across different X-ray energies, RSoXS contrast variation reveals environment-dependent mechanical behavior. Upon releasing strain, the X shape rotates and intensifies along the SD, likely due to the

formation of wrinkles, which introduce vacuum/roughness contrast dominance (Supplementary Fig. 45).

We define the angular positions of the X shape using the azimuthal angles, $\varphi_1$ and $\varphi_2$, where $\Delta\varphi$ (= $\varphi_2 - \varphi_1$) represents the degree of orientation relative to the SD (Fig. 3f, Supplementary Figs. 46, 47). As shown in Fig. 3g, h, $\Delta\varphi$ was first calculated at 3.54% strain, suggesting that elastic structural changes by minimal molecular rearrangements are limited to very low strain. Then, $\Delta\varphi$ shows a two-step growth, with a transition strain at 14.69%. In the first stage, $\Delta\varphi$ increases quickly, driven by initial chain orientation and rapid crystallite destruction, an irreversible process that provides an early, foundational orientation to the morphology. In the second stage, $\Delta\varphi$ displays another rapid increase, facilitated by the continued orientation and more chains freed by crystallite destruction, coupled with significant intrachain torsion at higher strains. Together, this sequential orientation process highlights a unique morphological adaptation in semiflexible conjugated polymers under mechanical deformation.

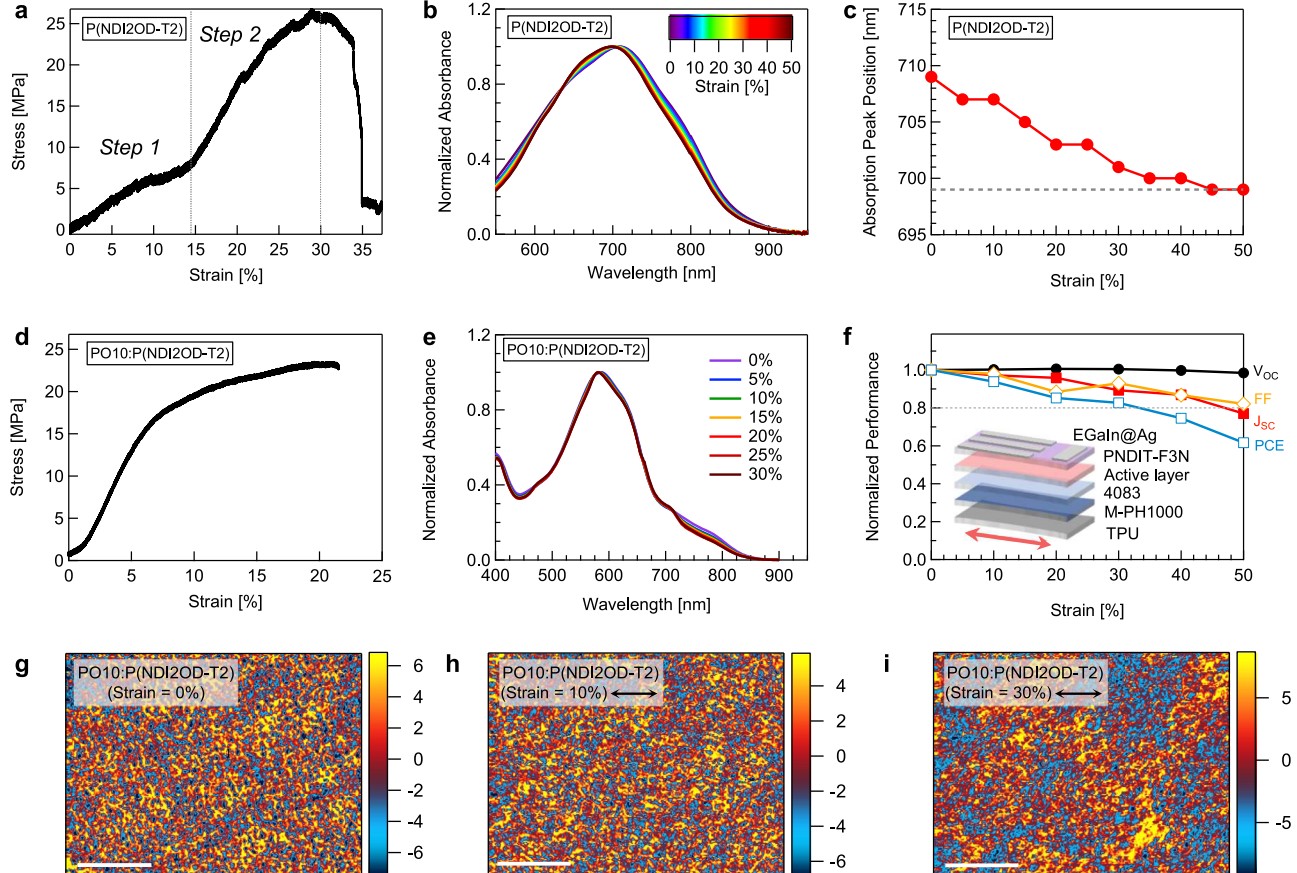

**Fig. 4 | Structure-property relationship analysis. a** σ-ε curve of the P(NDI2OD-T2) thin film. **b** UV-vis-NIR absorption spectra of the P(NDI2OD-T2) thin film during stretching with a film-on-elastomer sample. **c** Changes in the absorption peak position of the P(NDI2OD-T2) thin film during stretching. **d** σ-ε curve of the PO10:P(NDI2OD-T2) thin film. **e** UV-vis-NIR absorption spectra of the PO10:P(N-DI2OD-T2) blend film during stretching with a film-on-elastomer sample.

**f** Normalized device parameters of the PO10:P(NDI2OD-T2) based intrinsically stretchable OPV during stretching. AFM-IR images of the PO10:P(NDI2OD-T2) blend films at (**g**) 0%, (**h**) 10%, and (**i**) 30% strains, which highlight the P(NDI2OD-T2) domains with 843 cm⁻¹. The black double arrows indicate the stretch direction. The scale bar represents 0.5 μm, and the color bar corresponds to the IR amplitude.

Upon release, Δφ shows a linear decrease from 30% to 20.27% strain, after which it becomes relatively stable. The linear decrease is due to partial recovery of $\varphi_1$ and $\varphi_2$, indicating that chain orientations relax away from the SD. Notably, the Δφ at 20.27% aligns closely with that at 14.69% during stretch, suggesting that the structural relaxation involves the recovery of intrachain torsion, which is significantly enhanced during the later stage of stretching. However, beyond 20.27% strain, the stable Δφ suggests limited further relaxation, likely due to the formation of wrinkles that dissipate part of the applied strain energy (Supplementary Fig. 48)[46]. These results highlight the role of aligned chains with intrachain torsion across morphological scale for effective reversible stress dissipation.

## Correlations between structural evolution and properties

We further sought to connect the strain-induced multiscale structural changes with the mechanical, optical, and electronic properties of the thin film. As shown in Fig. 4a, the stress-strain (σ-ε) curve of the P(NDI2OD-T2) thin film displays an elastic region (~3%), a yield point (~10%), strain hardening (~14%), and ultimate fracture (~35%). Note that the crack-onset strain from optical microscopy (supported film) is not directly comparable to the fracture strain from tensile testing (free-standing film), as the former reflects delayed surface cracking under strain relaxation by the substrate, whereas the latter represents complete mechanical failure of the film. The yielding and hardening behaviors correspond well with the two-step Δφ growth observed in morphological adaptation under strain. In the initial stage, rapid

crystallite destruction and partial chain orientation contribute to morphological plasticity, distributing stress evenly across the film and preventing localized stress concentration, and thus leading to ductile yielding. In the subsequent stage, further chain alignment and the development of intrachain torsion over a large morphological scale increase resistance to additional strain, while effectively dissipating stress, thus contributing to the toughening and strain-hardening response. This connection underscores how multiscale structural changes in conjugated polymers translate directly to enhanced macroscopic mechanical properties, which helps understand why incorporating non-fused units and increasing molecular weight are effective chemical design methods for achieving stretchable conjugated polymers[47–51].

Next, we studied the UV-vis-NIR absorption spectra of the thin film during tensile stretching with a film-on-elastomer sample (Fig. 4b). Upon stretching to 50% strain, the main absorption peak at 709 nm exhibits a 10 nm blue shift, while the shoulder around 790 nm diminishes (Fig. 4c). This shift is attributed to the combined effects of crystallite destruction and intrachain torsion under strain, leading to a local order-disorder transition. Time-dependent density functional theory calculations using strain-dependent dimer structures confirm that increasing the dihedral angle between BT and NDI units from ~30° to 90° induces a blue shift in absorption (Supplementary Fig. 49), consistent with reduced π-conjugation and exciton delocalization. The absorption peak also narrows, which we attribute to increasing chain alignment across the morphological scale, limiting the distribution of

vertical transition energy levels. These findings highlight how multi-scale structural adaptations influence light-harvesting properties, which are critical for devices, such as organic photovoltaics (OPVs) and organic photodetectors (OPDs).

Intrinsically stretchable OPV devices with the structure of TPU/PEDOT:PSS (M-PH1000)/PEDOT:PSS (4083)/active layer/PNDIT-F3N/EGaIn@Ag were fabricated[52]. The active layer comprised the two conjugated polymers, PO10 (donor) and P(NDI2OD-T2) (acceptor), with a 2:1 weight blend ratio (Supplementary Fig. 50)[43]. This blend film shows ductile deformation, with a fracture strain of 21% observed from its σ-ε curve and crack-onset strain at 30% on PDMS substrate observed by optical microscopy (Fig. 4d, Supplementary Fig. 51). Similar to the neat thin film, the P(NDI2OD-T2) component in the blend shows blue-shifted absorption, during stretch from 0% to 30% strain, while PO10 contributes only minor changes within its own absorption range (Fig. 4e and Supplementary Fig. 52). The pre-stretched device exhibits a PCE of 6.92%, with a $V_{OC}$ of 0.854 V, a $J_{SC}$ of 13.97 mA cm$^{-2}$, and an FF of 58.00% (Supplementary Fig. 53, Supplementary Table 3). Upon stretching to 30%, the PCE drops to 5.78% ($V_{OC}$ = 0.858 V, $J_{SC}$ = 12.48 mA cm$^{-2}$, FF = 53.97%), indicating a loss of 84% of the initial PCE. Such decline is primarily due to the decrease in both $J_{SC}$ and FF (Fig. 4f), which correlate with charge generation, transport, and recombination in the devices.

The exciton dissociation and charge collection efficiency, $P(E,T)$, extracted from photocurrent density-effective voltage ($J_{ph}$-$V_{eff}$) curves, decreases from 98.86% for pre-stretched condition to 97.93%, 96.85%, and 95.53% at 10%, 20%, and 30% strain, respectively, indicating reduced charge generation (Supplementary Fig. 54). Photo-CELIV analysis reveals a concurrent decline in charge mobility from $6.59 \times 10^{-5}$ to $4.13 \times 10^{-5}$ cm$^2$ V$^{-1}$ s$^{-1}$ (Supplementary Fig. 55). The light intensity ($P_{light}$) dependence of $V_{OC}$, described by $V_{OC} \propto n \frac{kT}{q} \ln(P_{light})$, yields n values of 1.25, 1.42, 1.46, and 1.58 as strain increased, signifying enhanced trap-assisted recombination (Supplementary Fig. 56). To further elucidate this, morphological changes upon thin film stretched by nano-infrared spectroscopy atomic force microscopy (AFM-IR). As shown in Fig. 4g–i, P(NDI2OD-T2) domains probed with 843 cm$^{-1}$ exhibit nanofibrillar network morphology in the as-cast blend film (Supplementary Fig. 57), which is retained when 10% strain was applied, but transformed into large aggregates at 30% strain. This morphological transformation arises from strain-induced chain orientation, intrachain torsion, and crystalline destruction, leading to local order-disorder transitions. These combined absorption, electrical, and morphological results demonstrate that strain-induced structural evolution directly deteriorates charge generation and transport while promoting recombination losses, ultimately accounting for the observed performance degradation. To assess the effect of cyclic strain, PO10:P(NDI2OD-T2) films on PDMS were stretched at 20% for 300 cycles. AFM images show that the fibrillar morphology gradually blurred with repeated cycling (Supplementary Fig. 58), causing some decline in device performance; however, the device retained over 80% of its initial PCE (Supplementary Fig. 59 and Supplementary Table 4). These results indicate that the overall nanostructure and device performance remain largely preserved under moderate morphological relaxation.

Therefore, these results underscore that strain-induced multiscale structural changes inevitably compromise some electronic properties, a critical consideration for stretchable organic electronics. Recent studies have identified entropic chain alignment and enthalpic crystallite breakage as multimodal energy dissipation pathways for mechanical robustness[19]. Both effects are directly observed in our system, with a strain-dependent temporal evolution further revealed. Consistent with a recent review highlighting the role of chain alignment and torsional rotation in mechanical-electronic coupling, the in situ and simulation data show increased torsional freedom under strain, likely governed by backbone architecture[53]. Our findings suggest that incorporating additional stress dissipation mechanisms, such as dynamic bonding and elastomer networks[19,48,52–57], can mitigate structural deformation and improve device performance.

## Discussion

By integrating NEXAFS, TReXS, and in situ RSoXS, we achieve a correlative, multiscale analysis of structural evolution in conjugated polymer thin films under tensile deformation. This multimodal approach uncovers a previously unreported two-step response: an initial phase of polymer chain orientation and rapid crystallite fragmentation, followed by further chain alignment accompanied by intrachain torsional deformation at higher strains. These sequential structural adaptations collectively enable efficient stress dissipation (Fig. 5). Such changes play a pivotal role in modulating the mechanical properties of the films and directly influence both their optical absorption and photovoltaic performance.

Although the two-step morphological response is demonstrated using the highly crystalline P(NDI2OD-T2), its occurrence is expected to depend on the molecular structure and thin-film morphology. Ongoing studies on polymers with lower crystallinity or near-amorphous states aim to extend these insights and examine the generality of the observed behavior. Nevertheless, our findings establish a mechanistic framework for understanding how conjugated polymer systems maintain device functionality under large deformations. This insight provides a foundation for the rational design of materials that combine mechanical resilience with high optoelectronic performance. Notably, the structural mechanisms identified here are inaccessible to conventional ex situ or single-scale characterization techniques, highlighting the critical role of in situ RSoXS for real-time, multiscale structural analysis under operational strain conditions. Together, this work opens a new regime of structural-mechanical-functional correlation in organic electronics and provides a blueprint for next-generation characterization tools and materials design strategies in intrinsically stretchable electronics.

## Methods

### Sample preparation and general characterizations

P(NDI2OD-T2) was synthesized according to a reported procedure[58]. Its molecular weight was characterized by high-temperature gel permeation chromatography (GPC) on a Tosoh EcoSEC HT system equipped with three TSKgel GMHhr-H(S) HT columns in series, using 1,2,4-trichlorobenzene as the eluent at 135 °C. Calibration was carried out against narrow-dispersity (Đ) polystyrene standards. The number-average molecular weight ($M_n$) and dispersity of P(NDI2OD-T2) were determined to be 131 kDa and 1.356, respectively. The corresponding molecular-weight distribution curve is shown in Supplementary Fig. 60. The polymer PO10 was synthesized following a reported protocol[50], with $M_n$ = 24.4 kDa and Đ = 2.15. The $M_n$ of PNDIT-F3N was determined to be 12.3 kDa by GPC (Waters 2410) using tetrahydrofuran (THF) as the eluent and linear polystyrene standards for calibration. Preparation of thin films of P(NDI2OD-T2) involved several steps: (1) the PEDOT:PSS (Clevios P VP AI 4083, Heraeus) was spin-coated on a wafer at 4000 rpm; (2) P(NDI2OD-T2) thin films (~200 nm thick) were prepared by spin-coating on the PEDOT:PSS/silicon wafer substrates from a chloroform solution (10 mg mL$^{-1}$). The thicknesses of films were measured by a Tencor Alpha-step 500 Surface Profilometer. The film was then treated with thermal annealing at 100 °C on a hotplate for 10 min. The PDMS membrane was prepared by mixing 6 g base and 0.5 g cross-linker in a 20 mL vial with weight ratio of 12:1 (Sylgard 184, Dow Corning). The mixture was placed in a vacuum box for 10 min to remove the bubbles and then transferred on a 4-inch silicon wafer, and baked in a 70 °C oven for 1 h. Both film-on-elastomer and free-standing samples were prepared following the method as illustrated in Supplementary Fig. 3.

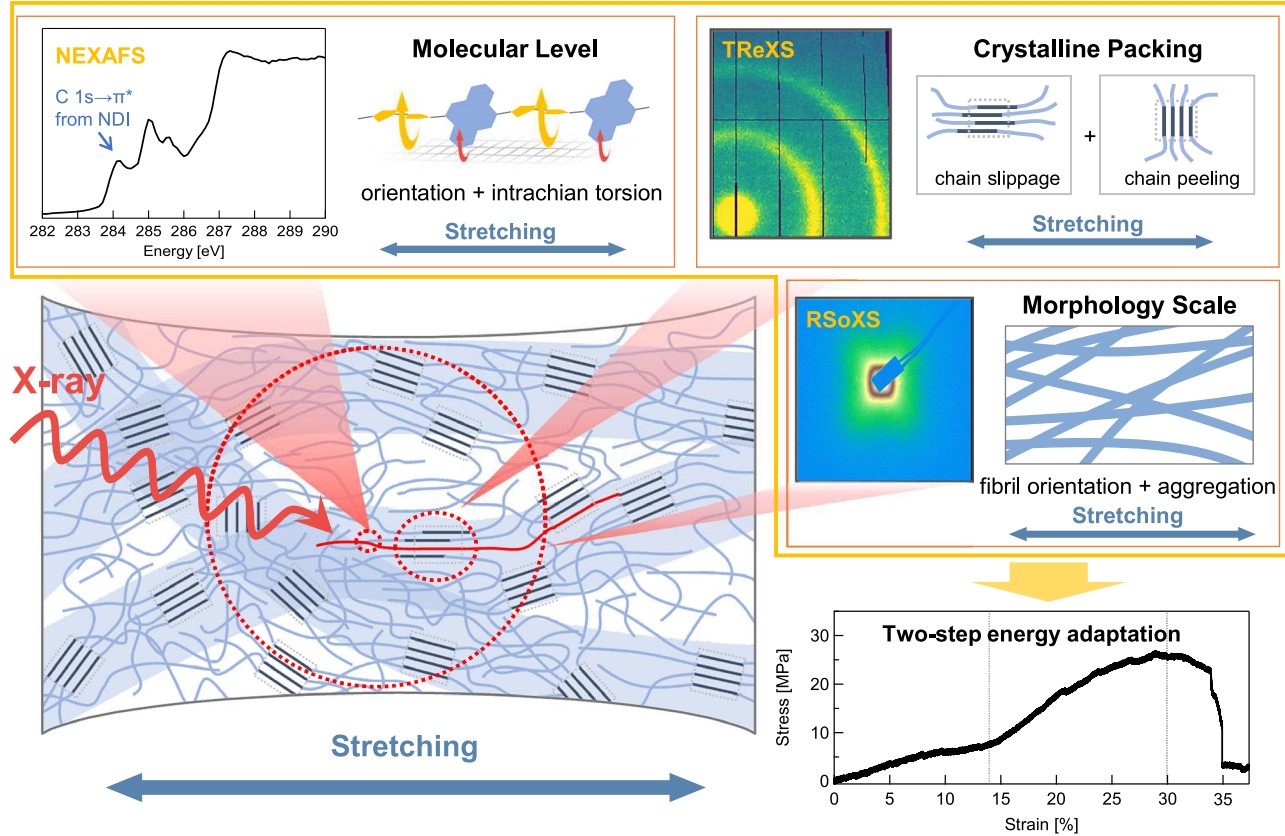

**Fig. 5 | Correlative multiscale structural evolution under tensile strain.** The molecular-to-mesoscale structural evolution in thin film of P(NDI2OD-T2), probed by a suite of X-ray techniques (NEXAFS, TReXS, and RSoXS), elucidates the two-step stress dissipation mechanism.

UV-vis-NIR absorption spectra of thin films were measured by a SHIMADZU UV-3600 spectrophotometer. The film-on-elastomer sample was placed with the clamps of a tensile stage and then subjected to the desired strains for the measurements. A pristine PDMS membrane with the same thickness as the film sample was used as a reference to subtract the absorption background. The resulting spectra were normalized to compare their differences in wavelength shift. AFM height images of the PNDI2OD-T2 thin films were captured by a Bruker Multimode 8 Atomic Force Microscope. AFM-infrared (AFM-IR) spectroscopy was performed using an Anasys nanoIR3 system (Bruker). The thin films were stretched to specific strains and transferred onto silicon wafers for imaging.

## NEXAFS simulations
The NEXAFS simulations of P(NDI2OD-T2) were done with its simplified structural unit BT-NDI, whose geometry was optimized with density function theory (DFT) at the B3LYP/6-31G* level using the Gaussian 09 package. The resulting molecular model was then used as input for NEXAFS simulations based on the excited electron and core hole (XCH) calculations. The resulting spectra were shifted by 284.5 eV for C K-edge. However, the PBE-GGA functional that is employed can underestimate the bandgap, leading to compressed spectra at the energy axis. To alleviate this issue, a dilation factor of 1.1 was applied to C K-edge simulations. The resulting transitions were converted to continuous spectra, which were more relevant to experimental data by using a Gaussian function with peak widths of 0.2 eV. We note that despite these corrections, small residual discrepancy of ~0.2–0.3 eV remains, as the simplified molecular model cannot fully capture the long-range conjugation, and the vacuum-based calculations cannot account for solid-state interactions. These deviations are commonly accepted in DFT-based NEXAFS

simulations and do not affect the reliability of spectral assignments[33,36,59].

## MD simulations
The MD simulation to study the dihedral angle changes of P(NDI2OD-T2) under stretching were performed as follows: (1) The structural geometry of a P(NDI2OD-T2) dimer was optimized using density functional theory (DFT) calculations at the B3LYP/6-31g** level, implemented in Gaussian 16[60] and analyzed with Multiwfn software[61]. The optimized dimer structure was then used as the repeat unit to construct a decamer chain of the polymer. (2) An amorphous cell containing the decamer chains was built with an initial density of 0.3 g cm$^{-3}$ to ensure dense packing. (3) The cell was subjected to incremental applied stress along the $x$ axis (0.001, 0.05, 0.10, 0.15, 0.16, 0.17, 0.18, 0.19, 0.20, 0.21, 0.22, and 0.23 GPa). For each stress, 100 ps of MD simulations were performed under the NVT ensemble at 300 K to capture the evolution of dihedral angles between specific units. The absorption spectra were calculated using a Gaussian 16 software based on cam-B3LYP-D3/TZVP, upon a tetramer structure extracted from the stress-strain MD simulation with rising torsion angle.

## TReXS
The TReXS measurements were performed at the Soft Matter Interfaces (SMI) beamline (Beamline 12-ID) of the National Synchrotron Light Source II, Brookhaven National Laboratory. The P(NDI2OD-T2) thin films were transferred onto PDMS substrates with a central circular hole (4 mm in diameter), as illustrated in Supplementary Fig. 3b. The X-ray beam size was 20 μm × 200 μm, ensuring that the beam fully passed through the unsupported film region. The sample was mounted in a tensile stage (Linkam TST350) in vacuum. Both static and in situ scattering images were collected by a Pilatus 300K-W detector

under a photon energy range of 2445–2500 eV, with a refined step size of 0.25 eV for the data acquisition through S K-edge (2470–2480 eV). Under TReXS, the reflection intensity is proportional to the square of the structure factor by $I(\boldsymbol{q}) \propto |F(\boldsymbol{q})|^2$. The structure factor, $F(\boldsymbol{q})$, is calculated using the equation $F(\boldsymbol{q}) = \sum_{j}^{atoms} f_j \exp(2\pi i \boldsymbol{q} \cdot \boldsymbol{r}_j)$, where $\boldsymbol{r}_j$ represents the location of the $j_{th}$ atom in the crystal unit cell and $f_j$ represents the energy-dependent atomic scattering factor of the $j_{th}$ atom. Changes in reflection intensity at resonance are caused by variation the $f_j$, which has a real component and an imaginary component, denoted as $f = f_0 + f' + if''$. The imagery part, $f''$, is determined using NEXAFS spectra, while the real part, $f'$, is obtained through a Kramers−Kronig transformation. To ensure consistent comparison across all strain levels, the scattering intensity was normalized to the value at q = 0.37 Å$^{-1}$ at 0% strain. No diffraction contribution from PDMS was observed; PDMS typically shows a reflection at q-0.40 Å$^{-1}$ (Supplementary Fig. 61), whereas our TReXS data remain flat in this range, confirming that the measured scattering originates solely from the polymer thin film.

## GIWAXS
The GIWAXS measurements of static stretched films were performed at beamline 7.3.3 of ALS, LBNL. The beam energy was 10 keV. The samples were placed in a Helium box with a sample-to-detector distance of ~280 mm, which was calibrated by a silver behenate. The incident angle was 0.16°, which was normalized by a photodiode. The scattering images were collected by a CCD detector (Pilatus 2M) with a pixel size of 0.172 mm × 0.172 mm and an exposure time of 2 s.

## NEXAFS and RSoXS
The uniaxial tensile tests of the thin film samples were performed by a tensile stage (200 N, Deben UK Ltd). The elongation and movement rate of clamps were controlled and monitored by a controller that was coupled with *Microtest* v6.3.30, a computer-based software. NEXAFS (TFY mode) and RSoXS were performed at beamline 11.0.1.2 of ALS, LBNL. The CCD detector (Princeton Instrument PI-MTE) has a pixel size of 0.027 mm × 0.027 mm. For NEXAFS, the tensile rate was set at 1.5 mm min$^{-1}$. Post-edge normalization of NEXAFS spectra was done by *Athena* 0.9.26. For the calculation of refractive indices, the absorptive component β was determined from NEXAFS spectra by β = μ/2k, where μ denotes the attenuation coefficient and k is the wave vector. The dispersive component δ was then obtained via Kramers-Kronig transformation with β. For in situ RSoXS, the tensile rate was 0.5 mm min$^{-1}$. The sample preparation followed the same procedure as shown in Supplementary Fig. 3b. The beam size is 200 μm × 200 μm, ensuring that the beam fully passed through the unsupported film region. The sample-to-detector distance was set at 150 mm. During the tensile test, the scattering images were captured consecutively by the CCD with 5 s exposure time per image.

## Stress-strain (σ-ε) curves of thin films
Tensile tests were conducted using a free-standing sample following the reported method[50]. The polymer solution was spin-coated onto a PEDOT:PSS/silicon wafer substrate. The sample was then floated on deionized water, allowing the PEDOT:PSS layer to dissolve and leaving the active layer free-standing on the water surface. Frosted aluminum blocks attached to the clamps of a tensile stage were carefully adjusted to capture the floating film via van der Waals interactions. After securing the film, the deionized water was gradually removed, leaving the active layer suspended on the tensile stage. The film was stretched at a constant rate of 0.03 mm/min. Stress (σ) and strain (ε) were calculated using the following equations: σ = F/(A×B) and ε = Δl/l$_0$, where *F* is the applied force measured by a high-sensitivity force sensor, *A* is the film width (10 mm), *B* is the film thickness, Δ*l* is the elongation of the film, and *l*0 is the initial distance between the clamps.

## Fabrication of intrinsically stretchable OPV devices
The stretchable OPV devices were fabricated with the structure TPU/PEDOT:PSS (M- PH1000)/PEDOT:PSS (4083)/active layer/PNDIT-F3N/EGaIn@Ag. The stretchable, transparent electrode was prepared by modifying PEDOT:PSS (PH1000, Heraeus) with the addition of 5 vol% dimethyl sulfoxide, 2 vol% polyethylene glycol, 0.5 vol% FS-30, and 0.1% GOPS. The resulting solution was spin-coated onto a TPU substrate at 1500 rpm, followed by baking at 100°C for 20 min. Next, PEDOT:PSS (4083, Heraeus) was spin-coated onto the PH1000/TPU layer at 3000 rpm and dried at 100 °C for 20 min. The PO10:P(NDI2OD-T2) blend film (2:1 weight ratio) was then spin-coated with its chloroform solution (6 mg/mL for PO10, with 0.5 vol% dibenzyl ether as solvent additive) onto the 4083 layer as the active layer. Post-treatment to the active layer with thermal annealing of 100 °C for 10 min. Then, PNDIT-F3N (methanol solution, 1 mg/mL) was spin-coated at 3000 rpm on the active layers. Finally, a 100 nm thick Ag layer was thermally evaporated onto the interface through a shadow mask in a vacuum chamber at a pressure of $3 \times 10^{-7}$ torr. EGaIn liquid metal was then sprayed onto the Ag layer using a deposition mask, yielding the EGaIn@Ag stretchable electrodes. The initial active area (0.04 cm$^2$) was defined by the overlap of the patterned M-PH1000 bottom electrode and the patterned EGaIn@Ag top electrode in a cross-electrode geometry. During mechanical stretching, the active area deformed geometrically. To account for this, the actual active area was corrected based on dimensional changes measured by optical microscopy. The average area variation was extracted from three independent devices under identical strain conditions and fitted with a three-phase exponential decay function (Supplementary Fig. 62), which was then applied to correct the active area of the tested devices at each strain. *J-V* curves were measured using a Keithley 2400 source meter with illumination (100 mW cm$^{-2}$) provided by an AM 1.5G solar simulator (SS-F5-3A, Enlitech) in a nitrogen-filled glovebox. Photo-CELIV measurements were performed using the PAIOS platform (FLUXiM, Switzerland).

## Reporting summary
Further information on research design is available in the Nature Portfolio Reporting Summary linked to this article.

# Data availability
All data supporting the findings of this study are included in the manuscript or the Supplementary Information. Source data are provided with this paper.

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

## Acknowledgements

This work was supported by the National Key R&D Program of China (2023YFB3609000 to W.Z.), National Natural Science Foundation of China (22109094 to W.Z.; 52325306 to F.L.), China Postdoctoral Science Foundation (2022M712054 to W.Z.), and State Key Lab of Luminescent Materials and Devices, South China University of Technology (Skllmd-2025-09 to W.Z.). RSoXS was done at beamline 11.0.1.2 at the Advanced Light Source, Lawrence Berkeley National Laboratory (LBNL). The computational methods were supported by a User Project at The Molecular Foundry at LBNL. Supercomputer time was provided by the Lawrencium computational cluster administered by the High-Performance Computing Services Group at LBNL. All LBNL efforts were supported by the Office of Science of the U.S. Department of Energy under Contract no. DE-AC02-05CH11231. TReXS was performed at the Soft Matter Interfaces (SMI) beamline (Beamline 12-ID) of the National Synchrotron Light Source II, a U.S. Department of Energy (DOE) Office of Science User Facility operated for the DOE Office of Science by Brookhaven National Laboratory under Contract DE-SC0012704.

## Author contributions

W.Z., C.W., and F.L. conceived and designed the project. W.Z. performed formal analysis, methodology development, and validation; led the investigation, funding acquisition, project administration, and original draft writing; and contributed to supervision and review/editing of the manuscript. G.F. contributed to investigation and provided resources. G.M.S. supported investigation and contributed to manuscript review/editing. S.W., X. Liu, W.Y., L.Y., and X.W. conducted supporting investigations. X. Luo and Y. L. contributed to formal analysis. T.J.F. contributed to manuscript review and editing. T.P.R. provided supporting resources and contributed to review/editing. L.Y., F.H., and Y.Z. provided supporting resources. C.W. and F.L. led project conceptualization, administration, and supervision, supported resource acquisition, and contributed to original draft writing and editing.

## Competing interests

The authors declare no competing Interests.
