## [Transparent Peer Review file · Nature Communications]

Correlative molecular-to-mesoscale evolution in conjugated polymers for intrinsically stretchable organic photovoltaics

Corresponding Author: Professor Feng Liu

Version 0:

Reviewer comments:

Reviewer #1

(Remarks to the Author)

The manuscript by Zhong et al. combined NEXAFS, TReXS, and in situ RSoXS to study the microscale to mesoscale structural evolution of a model conjugated polymer P(NDI2OD-T2) under mechanical strain. The study reveals a two-step response mechanism that finally causes the crystal peeling effect under large strain: first, the stage of polymer chain orientation and rapid crystallite fracture, and secondly, the stage of chain alignment and intrachain torsional deformation. Further, the authors studied P(NDI2OD-T2):PO10 blend systems and makes a correlation between device behavior and morphological disruption resulting from strain-induced meso- and microscale crystalline destruction investigated from resonant X-ray scatterings. With minor revisions and more experiments, this paper is well-suited for publication in Nature Communications:

1. In lines 117 to 122, the authors used different mathematical equations to fit the tilt angle of the transition dipole moment under 0% and 50% strain. It seems the expression used for the 0% strain condition ($I = A/3[1+12(3\cos^2\theta-1)(3\cos^2\alpha-1)]$) is assuming a uniaxial orientation distribution where the stacking is face-on predominantly in the z-direction but in-plane orientation in the x and y directions is random. While this may be a reasonable assumption for an unstressed system, there are some conjugated polymers that have some in-plane alignment before stretching depending on the film forming process. The authors should clarify whether this assumption is supported by earlier statements in the manuscript, specifically the claim that β_x equals β_y at 0% strain, and whether such a model is universally valid for the polymer system under study.
2. In line 132, the authors refer to "Area 1" in the NEXAFS analysis, but the methodology used to extract this value is not described in detail. I assumed that the area under the curve was calculated directly from a fixed energy window. However, the authors should discuss in SI or method part whether a peak fitting approach was considered, and whether this method might offer a more accurate representation of the electronic transitions. Authors should also mention, if no fitting is performed, if using such a peak fitting would change the observed trend or only shift the numerical values.
3. The display of figure 2 is confusing. First, the italicized terms "para." and "perp." overlap with the labels for crystal directions, which create confusion for the reader. Replacing these overlapping labels with clearly marked axes or direction arrows would enhance readability. Second, the data shown in Figures 2c to e would be more interpretable if the corresponding scattering features were marked using the same color to represent. For example, the author can use four different colors (grouped into two main group, like deep and light shades of blue and red) and denote the corresponding scattering feature and also use the corresponding color in figure 2c (instead of plain yellow).
4. The authors have discussed the scattering pattern at 2450 eV in TReXS but had not offered further discussion. Given that this energy also excites the (100) peak, it would be useful to know whether the (100) scattering shows the same decay profile under strain as excited by other energy. If the behavior differs, this may suggest distinct mechanical responses in the polymer chain in different environments. The authors should include it in either SI or the body.
5. In line 225, the authors argue that the RSoXS results align with AFM data; however, AFM images are only provided for samples strained to 50%, which is significantly larger than that of the RSoXS experiment (30%). It remains unclear at what strain level the "X-shaped" anisotropy emerges in AFM. Including an FFT analysis of AFM images at intermediate strains, such as around 30%, would help confirm the structural evolution pathway supported by the X-ray scattering data. Additionally, a comparison of domain sizes derived from AFM and RSoXS would be valuable. Although the feature is not seemingly a peak, the authors still may select some turning points in the small angle range if plot a figure on log-log scale.
6. The UV-Vis spectrum of the blend film is unclear. The spectral features that appear in the blend film is significantly shifted compared to neat P(NDI2OD-T2). It is unclear whether this shift originates from the P(NDI2OD-T2) component or from PO10. The authors should include a strain-dependent UV-Vis spectrum of neat PO10 films to determine the individual contributions of each material in the blend. This would allow them to confidently attribute the observed blue shift in the blend to a specific

molecular origin.

7. While the manuscript connects morphological evolution to device performance, the authors could elaborate further on how the relatively subtle strain-dependent optical changes (only a few nm shifts in absorption peaks) influence charge generation, transport, or recombination in OPV devices.

8. The authors could expand the discussion of prior works on strain-induced morphology changes in conjugated polymers. For example, Wu et al. (<https://www.nature.com/articles/s41467-023-44099-w>) introduced the “relative stretchability” (rS, the ratio of DR and rDoC) to quantitatively analyze the strain induced morphological evolution and crystal breakage. Comments on whether your system follows similar entropic (chain alignment, from DR) and enthalpic (crystallite amorphization/breakage, from rDoC) dissipation pathways would be helpful for the readers. In addition, Zheng et al. (<https://pubs.acs.org/doi/full/10.1021/jacs.2c00072>) summarized the molecular and morphological processes expected under strain, for example, chain alignment and strain-induced backbone planarization through monomer torsional rotation and then outlined how such processes map onto mobility retention. Since this manuscript shows direct evidence on monomer rotation to planarize the chain under strain, highlighting how it supports or refines the assumptions in this review paper would be helpful.

Reviewer #2

(Remarks to the Author)

The manuscript presents a fundamental investigation into the structural evolution of a conjugated polymer under tensile stress, utilizing a combination of NEXAFS, TReXS, and RSoXS techniques. The studied material is further demonstrated in a fully stretchable photovoltaic device. Overall, the manuscript is well written, and the comprehensive suite of characterizations are clearly correlated to provide a coherent mesoscale picture that advances the fundamental understanding of intrinsically stretchable organic semiconductors during dynamic mechanical deformation. I recommend this work for publication, provided that the following concerns and questions are satisfactorily addressed:

Q1: In comparing the simulated NEXAFS spectra of P(NDI2OD-T2) with the experimental results in Figure 1c, there remain noticeable discrepancies, particularly around 285.1 eV. What might be the origin of these mismatches? Additionally, the major and minor tick marks on the x-axis of Figure 1c are not clearly visible and should be improved for clarity.

Q2: Regarding the free-standing experimental setup, as the majority of the film remains supported by the PDMS substrate, could this result in differences between the local strain experienced near the PDMS opening and the overall applied strain? How is the strain distributed across the film, and is there any quantification or mapping of potential strain gradients?

Q3: For the TReXS and RSoXS experimental setups, which employ a transmission geometry, how is it ensured that the X-ray beam does not also penetrate and contribute signal from the PDMS substrate? What is the beam size relative to the size of the unsupported region (hole)? Additionally, since the scattering intensity is correlated with film thickness—and considering that the film thickness is likely reduced during stretching—is this variation reflected or corrected for in the analysis (for example, in Figure 2d)?

Q4: While optical microscopy determined the crack-onset of P(NDI2OD-T2) to occur at strains above 50% (Fig. S2), the macroscopic stress-strain curve exhibits mechanical failure before reaching 35% strain (Fig. 4a). Can the authors clarify the origin of this apparent discrepancy?

Q5: The observation of a two-stage strain response corresponding to distinct stress dissipation mechanisms in P(NDI2OD-T2) is intriguing. To what extent is this behavior generalizable to other conjugated polymers, such as PO10, already studied in this work?

Q6: The demonstration of robust photovoltaic device performance under up to 50% applied strain is impressive. Is it possible to directly characterize the charge carrier mobility of P(NDI2OD-T2) under strain? Such measurements would provide important insight into the relationship between mechanical deformation, structural evolution, and charge transport properties.

Q7: For real-world applications, stretchable semiconductors will undergo repeated cyclic stretching rather than a single extension. How do the authors anticipate that cyclic mechanical fatigue will influence the nanostructure and electrical performance over time? Are there any preliminary results or plans to study such effects?

Q8: The molecular weight of the P(NDI2OD-T2) is very high, 131k; providing the P(NDI2OD-T2) GPC traces and the calibration curves will be helpful. Besides, material information (synthetic procedure, molecular weight) on PO10 and PNDIT-F3N is missing.

Reviewer #3

(Remarks to the Author)

This manuscript presents a novel and comprehensive approach to study the multiscale structural evolution of conjugated polymer thin films under strain, using a powerful combination of advanced in situ X-ray techniques (NEXAFS, TReXS, RSoXS). The study provides important mechanistic insights into stress-dissipation pathways, including chain orientation changes, intrachain torsion, crystallite disruption (via “peeling” and “slippage” mechanisms), and a “two-step” morphological reorganization. Such structural changes are successfully connected to mechanical and optoelectronic properties. The experimental design is rigorous, the correlative multimodal methodology is state-of-the-art, and the results offer valuable guidance for the rational design of durable stretchable electronics. Given the novelty of the characterization method and the

depth of the mechanistic insights, I strongly recommend publication after minor revisions.

(1) While the title emphasizes the study of structural evolution on organic photovoltaic, the Introduction does not clearly articulate how key morphological changes under strain impact photovoltaic metrics. A more explicit discussion connecting the multiscale structural changes to performance-limiting mechanisms in organic solar cells would enhance the impact of the work.

(2) The angle-resolved NEXAFS shows increased α -angles of polymer chains under strain. Are similar changes observed within the crystalline domains? Since face-on π - π stacking is crucial for vertical charge transport in OSCs, clarification on whether the crystallite orientation is similarly affected would be helpful.

(3) The observed decrease in crystallinity during stretching is reasonable due to the rigid backbone of conjugated polymers. However, it remains unclear why the scattering intensity continues to decrease upon strain release.

(4) The use of $\Delta\phi$ to quantify morphological orientation is well-executed and convincingly correlates with mechanical strain (σ - ϵ). However, the physical origin of the "X"-shaped contrast in the RSoXS images should be clarified due to the structural complexity of the stained thin film. Meanwhile, how are the ϕ positions determined?

(5) The manuscript suggests that crystallite disruption and increased torsion reduce exciton delocalization, affecting optical absorption. Do the authors have supporting evidence or citations for this claim?

(6) Regarding the J-V curves of stretchable solar cells, how is the device's active area defined and measured during strain application? This point should be clarified in the experimental section.

Reviewer #4

(Remarks to the Author)

Reviewer #5

(Remarks to the Author)

Version 1:

Reviewer comments:

Reviewer #1

(Remarks to the Author)

The authors have adequately addressed the manuscript revisions, and this version has improved significantly. Most of my comments have been addressed with additional experiments and explanations in the text where needed. The only part that remains slightly confusing is the additional AFM analysis for comment 5. The authors introduced power spectral density (PSD) curve analysis of the AFM results, but the results shown in Fig. S41 differ from their description. From the figure, the order of domain size from small to large is 0% (black), 10% (red), 30% (blue), and 20% (yellow), while in the figure caption it was listed as 0%, 10%, 20%, and 30%. Additionally, the authors did not apply the same PSD analysis to their original AFM data from the 50% strain film for comparison. While the manuscript can be published in Nature Communication, I would recommend some additional clarification of the newly added AFM analysis.

Reviewer #2

(Remarks to the Author)

The revision has addressed all the comments. I now recommend accepting the paper.

Reviewer #3

(Remarks to the Author)

The authors have thoroughly addressed my comments, and the manuscript has been revised accordingly. Thus, I would recommend it for publication.

Reviewer #4

(Remarks to the Author)

Reviewer #5

(Remarks to the Author)

Response to Reviewers' Comments

Reviewer #1 (Remarks to the Author):

The manuscript by Zhong et al. combined NEXAFS, TReXS, and in situ RSoXS to study the microscale to mesoscale structural evolution of a model conjugated polymer P(NDI2OD-T2) under mechanical strain. The study reveals a two-step response mechanism that finally causes the crystal peeling effect under large strain: first, the stage of polymer chain orientation and rapid crystallite fracture, and secondly, the stage of chain alignment and intrachain torsional deformation. Further, the authors studied P(NDI2OD-T2):PO10 blend systems and makes a correlation between device behavior and morphological disruption resulting from strain-induced meso- and microscale crystalline destruction investigated from resonant X-ray scatterings. With minor revisions and more experiments, this paper is well-suited for publication in Nature Communications:

1. In lines 117 to 122, the authors used different mathematical equations to fit the tilt angle of the transition dipole moment under 0% and 50% strain. It seems the expression used for the 0% strain condition ($I = A/3[1+1/2(3\cos 2\theta-1)(3\cos 2\alpha-1)]$) is assuming a uniaxial orientation distribution where the stacking is face-on predominantly in the z-direction but in-plane orientation in the x and y directions is random. While this may be a reasonable assumption for an unstressed system, there are some conjugated polymers that have some in-plane alignment before stretching depending on the film forming process. The authors should clarify whether this assumption is supported by earlier statements in the manuscript, specifically the claim that β_x equals β_y at 0% strain, and whether such a model is universally valid for the polymer system under study.

Our response: Thanks for the comment. At 0% strain, we adopted the fitting equation commonly used for samples with three-fold or higher substrate symmetry. The as-cast P(NDI2OD-T2) thin films were prepared by spin-coating without any external alignment field, and thus in-plane isotropy is expected. We agree with the reviewer that this assumption is not universally valid for all conjugated polymer systems, since

certain systems can indeed exhibit in-plane alignment depending on the film-forming process. To support our assumption in the present case, we added GIWAXS measurements of the as-cast P(NDI2OD-T2) thin film probed with orthogonal X-ray directions. The results show nearly identical reflections with similar peak positions, intensities, and widths, which confirms that $\beta_x \approx \beta_y$ before stretching. In addition, we revised the manuscript to clarify that the fitting model at 0% strain is specifically justified for the current P(NDI2OD-T2) system and should not be considered a universal assumption for all conjugated polymers. The corresponding revisions are included in the revised manuscript.

Main text (Page 9, line 7):

“To study the π - π stacking developments during stretch, the films were stretched to the desired strains and then placed on silicon wafers for grazing incidence wide angle X-ray scattering (GIWAXS) measurements at the beam energy of 10 keV. This measurement was performed with orthogonal X-ray directions including parallel and perpendicular conditions (Fig. S28). The 0% strain thin film shows nearly identical reflections (Fig. S29), which support that $\beta_x \approx \beta_y$ before stretching as demonstrated by NEXAFS.”

Main text (Page 8, line 28):

“This angular dependence of NEXAFS intensity is appropriate for samples with three-fold or higher substrate symmetry, consistent with the as-cast P(NDI2OD-T2) thin films prepared by spin-coating. While some conjugated polymer systems can indeed exhibit IP alignment depending on the film-forming process ^{Error! Bookmark not defined.}, we note that this fitting method should not be considered universally valid for all thin films prior to stretching.”

Reference:

[39] Schuettfort, T., Thomsen, L. & McNeill, C.R. Observation of a Distinct Surface Molecular Orientation in Films of a High Mobility Conjugated Polymer. *J. Am. Chem. Soc.* **135**, 1092-1101 (2013).

Supplementary Information:

Fig. S29. GIWAXS (a,b) 2D images and (c) averaged I - q curves in IP and OOP directions of P(NDI2OD-T2) thin film under pre-stretched condition.

2. In line 132, the authors refer to “Area 1” in the NEXAFS analysis, but the methodology used to extract this value is not described in detail. I assumed that the area under the curve was calculated directly from a fixed energy window. However, the authors should discuss in SI or method part whether a peak fitting approach was considered, and whether this method might offer a more accurate representation of the electronic transitions. Authors should also mention, if no fitting is performed, if using such a peak fitting would change the observed trend or only shift the numerical values.

Our response: Thanks for the comment. In addition to calculating the integrated intensity from a fixed energy window, we also performed peak fitting of the corresponding NEXAFS features based on Gaussian functions. The results from the peak fitting analysis are consistent with those obtained by direct integration, showing the same trend in tilt angle changes under strain, with only minor numerical differences in the extracted values. To make this clearer, we have added a description of the fitting

details and results in the Supplementary Information and clarified that both approaches lead to the same conclusions.

Supplementary Information:

Fig. S11. Gaussian multi-peak fitting of the π^* region in angle-dependent NEXAFS spectra of P(NDI2OD-T2) thin films: (a-c) under 0% strain; (d-f) under 50% strain.

Fig. S12. Linear fitting of peak area of angle-dependent TFY NEXAFS spectra,

obtained from Gaussian multi-peak fitting of P(NDI2OD-T2) thin films at 0% and 50% strain. The parameters a and b correspond to the intercept and slope, respectively.

Fig. S13. Fitted tilt angles (α) derived from Gaussian multi-peak fitting for the peak at 284.2 eV and the total π^* region. The extracted α values for the 50% strain thin film are 34.6° and 35.2°, representing decreases of 6.4° and 9.5°, respectively, compared to the pre-stretched film. These results are consistent with those obtained from direct integration (Fig. S10), confirming the same trend in tilt angle reduction under strain, with only minor numerical differences.

3. The display of figure 2 is confusing. First, the italicized terms “para.” and “perp.” overlap with the labels for crystal directions, which create confusion for the reader. Replacing these overlapping labels with clearly marked axes or direction arrows would enhance readability. Second, the data shown in Figures 2c to e would be more interpretable if the corresponding scattering features were marked using the same color to represent. For example, the author can use four different colors (grouped into two main group, like deep and light shades of blue and red) and denote the corresponding scattering feature and also use the corresponding color in figure 2c (instead of plain yellow).

Our response: Thanks for the comment. To avoid overlap between the crystallite orientation labels and the q-vector directions, we have replaced “para.” and “perp.” with “q_x” and “q_y” to denote the respective q vectors. In addition, to improve the readability of Figures 2c-e, we now use two color groups: (i) deep and light shades of blue to

indicate the (001) and (100) reflections of crystallites aligned parallel to the stretch direction, and (ii) deep and light shades of red to indicate the (001) and (100) reflections of crystallites aligned perpendicular to the stretch direction. The revised Figure 2 is shown below.

Fig. 2. Crystallite deformation and destruction. (a) Experimental setup for TReXS and RSoXS in transmission geometry, where TReXS probes crystalline packing and RSoXS reveals the morphology of the P(NDI2OD-T2) thin films. (b) TReXS I - q curves averaged along the q_x and q_y directions at 2450 eV and 2476.26 eV. (c) Schematic illustrations of (100) and (001) reflections for parallel and perpendicular crystallites. (d) Intensity and packing number (CCL/ d -spacing) of the (100) and (001) reflections as functions of strain for parallel and perpendicular crystals.

4. The authors have discussed the scattering pattern at 2450 eV in TReXS but had not offered further discussion. Given that this energy also excites the (100) peak, it would be useful to know whether the (100) scattering shows the same decay profile under strain as excited by other energy. If the behavior differs, this may suggest distinct mechanical responses in the polymer chain in different environments. The authors should include it in either SI or the body.

Our response: Thanks for the comment. The (100) scattering decay profile under strain does not show significant differences between 2450 eV and other X-ray energies.

Specifically, GIWAXS measurements at 10 keV on strained films reveal that the (100) peak area in both q_x and q_y decreases rapidly at small strains and then more gradually at higher strains, with a stronger reduction along q_x . This behavior is consistent with the observations at 2476.26 eV in TReXS, indicating that the resonant excitation is not sensitive to the crystalline deformation. The data at 2450 eV were included to demonstrate that tender X-rays reproduce the same (100) and (001) reflection distributions as hard X-rays (10 keV), thereby validating the use of TReXS for probing strain-induced crystallite deformation. However, we did not perform in situ TReXS measurements at 2450 eV itself.

To further address the reviewer's point about environment-specific responses, we compared in situ RSoXS measurements at 284.2 eV and 285.0 eV, where the scattering contrast selectively probes different chemical environments. At 284.2 eV, stretching induces a clear "X"-shaped scattering pattern, reflecting fibril aggregate reorientation, whereas at 285.0 eV this feature is absent, suggesting reduced orientation sensitivity and a distinct response under strain. These results confirm that while the (100) crystalline scattering exhibits similar strain decay profiles across different X-ray energies, RSoXS contrast variation reveals environment-dependent mechanical behavior. We have added clarifying sentences to the manuscript accordingly.

Main text (Page 9, line 7):

"To study the π - π stacking developments during stretch, the films were stretched to the desired strains and then placed on silicon wafers for grazing incidence wide angle X-ray scattering (GIWAXS) measurements at the beam energy of 10 keV. This measurement was performed with orthogonal X-ray directions including parallel and perpendicular conditions (Fig. S28). The 0% strain thin film shows nearly identical reflections (Fig. S29), which support that $\beta_x \approx \beta_y$ before stretching as demonstrated by NEXAFS. Meanwhile, the (100) peak area in both q_x and q_y decreases rapidly at small strains (10-20%) and then more gradually at higher strains, with a stronger reduction along q_x (Fig. S30-32, Table S1-S2). This behavior is consistent with the observations at 2476.26 eV in TReXS, indicating that the resonant excitation is not sensitive to the

crystalline deformation”

Main text (Page 11, line 19):

“*In situ* RSoXS measurements at 285.0 eV during stretching show that “X”-shaped feature disappears (Fig. S44), suggesting reduced orientation sensitivity and further supporting the assignment of scattering contrast origin. Thus, the “X” pattern provides direct information related to the orientation of fibril aggregates. Such results also indicates that although the crystalline scattering exhibits similar strain decay profiles across different X-ray energies, RSoXS contrast variation reveals environment-dependent mechanical behavior.”

Supplementary Information:

Fig. S28. (a) Schematic representation of GIWAXS geometries for strained P(NDI2OD-T2) thin film, where the scattering images are collected with the incident X-ray roughly perpendicular (perp.) or parallel (para.) to the SD. (b) GIWAXS patterns of P(NDI2OD-T2) thin films with various strains measured with the incident X-ray beam (b) perpendicular (q_x) and (c) parallel (q_y) to the SD.

Fig. S30. Fitting illustration of GIWAXS (100) reflections in the q_x direction (X-ray beam perpendicular to the stretch direction) of P(NDI2OD-T2) thin films under different strains: (a) 0%, (b) 10%, (c) 20%, (d) 30%, (e) 40%, and (f) 50%.

Fig. S31. Fitting illustration of GIWAXS (100) reflections in the q_y direction (X-ray beam parallel to the stretch direction) of P(NDI2OD-T2) thin films under different strains: (a) 0%, (b) 10%, (c) 20%, (d) 30%, (e) 40%, and (f) 50%.

Fig. S32. GIWAXS (100) peak area in the q_x and q_y directions as a function of strain for P(NDI2OD-T2) thin films

Fig. S44 In situ RSoXS images of a P(NDI2OD-T2) thin film at different strains during (a) stretching and (b) releasing, which are probed at 285.0 eV.

Table S1. Fitted results of the GIWAXS (100) reflections in the q_x direction of P(NDI2OD-T2) thin film under different strains. FWHM represents full width at half maximum.

Strain [%]	Type	Location [\AA^{-1}]	Amplitude	Area	FWHM [\AA^{-1}]
0	Lorentzian	0.25	4545.02	180.37	0.0253
10	Lorentzian	0.25	3823.81	153.09	0.0255
20	Lorentzian	0.25	2492.22	107.47	0.0275
30	Lorentzian	0.25	2127.76	92.28	0.0276

40	Lorentzian	0.25	1299.00	60.51	0.0297
50	Lorentzian	0.25	982.19	46.52	0.0302

Table S2. Fitted results of the GIWAXS (100) reflections in the q_y direction of P(NDI2OD-T2) thin film under different strains. FWHM represents full width at half maximum.

Strain [%]	Type	Location [\AA^{-1}]	Amplitude	Area	FWHM [\AA^{-1}]
0	Lorentzian	0.25	4769.91	177.63	0.0260
10	Lorentzian	0.25	4014.66	153.62	0.0267
20	Lorentzian	0.25	3737.5	152.97	0.0296
30	Lorentzian	0.25	3643.62	152.61	0.0280
40	Lorentzian	0.25	3700.64	145.93	0.0285
50	Lorentzian	0.25	3705.69	145.55	0.0280

5. In line 225, the authors argue that the RSoXS results align with AFM data; however, AFM images are only provided for samples strained to 50%, which is significantly larger than that of the RSoXS experiment (30%). It remains unclear at what strain level the “X-shaped” anisotropy emerges in AFM. Including an FFT analysis of AFM images at intermediate strains, such as around 30%, would help confirm the structural evolution pathway supported by the X-ray scattering data. Additionally, a comparison of domain sizes derived from AFM and RSoXS would be valuable. Although the feature is not seemingly a peak, the authors still may select some turning points in the small angle range if plot a figure on log-log scale.

Our response: Thanks for the comment. To clarify the structural evolution with strain, we have now included additional AFM images at 0%, 10%, 20%, and 30% strain levels, along with their corresponding FFT analyses. In the as-cast film (0% strain), N2200 exhibits a fibrillar morphology that assembles into clusters, with slight anisotropy observable in the FFT. As strain increases, the fibrillar features become less distinct, and bundle-like domains grow in size. An “X”-shaped anisotropic signature begins to

emerge in the FFT around 20% strain, which becomes more pronounced by 30% strain. This progression aligns temporally with the development of the “X”-shaped scattering pattern observed in RSoXS, though the two techniques probe different structural levels with different sample configuration.

To quantify the morphology evolution, we extracted the power spectral density (PSD) from each AFM height image. The turning point in the log-log PSD plots provides an estimate of the characteristic domain size, which increased from 2.2 μm to 3.9 μm , 4.5 μm , and 4.8 μm for films strained at 0%, 10%, 20%, and 30%, respectively. These results reflect the micrometer-scale aggregation process captured by AFM.

RSoXS probes the nanometer sized features, which is the structure within the bundle domain. There is no turning point in I-q curves in the log-log scale. Thus, we analyzed the characteristic size with Debye-Bueche model, $(I(q))^{-1/2} = K(a^3Q)^{-1/2}(1 + a^2q^2)$, where K is a constant, Q the scattering invariant, and a the correlation length (CL). The CL initially decreased by about 50 nm at low strain (<5%), likely due to transverse compression, and then increased from ~150 nm to 185 nm as strain further increased. This trend reflects reorganization and alignment at the nanoscale, which complements the micro-scale aggregation observed by AFM.

Although AFM and RSoXS probe different length scales (micrometer vs. nanometer), both consistently reveal a strain-induced transition from fibrillar networks to globally oriented, aggregated domains. These complementary results support our conclusion that the anisotropic “X”-shaped scattering in RSoXS arises from the same fibrillar alignment observed by AFM. To clarify this, we added the following sentences in the manuscript.

Main text (page 10, line 15):

“For the 50% strained film, large-scale fibril aggregates with their long axes aligned toward the SD are seen, showing a fast Fourier transform (FFT) pattern with an “X” shaped anisotropy, which resulted from the progressive development in both size and orientation of the aggregates (Fig. S34). This reflects the formation of a statistically oriented structure induced by strain.”

Main text (page 11, line 12):

“The corresponding characteristic domain size was also evaluated using the Debye-Bueche model, which shows a trend of structural reorganization that complements the micro-scale evolution observed by AFM (Fig. S41-S43).”

Supplementary Information:

Fig. S34. AFM height images and corresponding FFT images of P(NDI2OD-T2) thin films under different stretch strains: (a) 0%, (b) 10%, (c) 20%, and (d) 30%. The stretch direction is in the horizontal direction of the images.

Fig. S41. Power spectral density (PSD) plots extracted from AFM height images (as shown in Fig. S34) of P(NDI2OD-T2) thin films under different stretch strains: (a) 0%, (b) 10%, (c) 20%, and (d) 30%. The dotted lines highlight the turning points giving an estimate of the characteristic domain size, which increased from $2.2 \mu\text{m}$ to $3.9 \mu\text{m}$, $4.5 \mu\text{m}$, and $4.8 \mu\text{m}$ for films strained at 0%, 10%, 20%, and 30%, respectively. These results reflect the micrometer-scale aggregation process captured by AFM.

Fig. S42. (a-b) The I - q curves, averaged at two azimuthal angles of the X-shaped signal in RSoXS, i.e., ϕ_1 (Sector 1) and ϕ_2 (Sector 2). Inset shows the sector averaging details of Sector 1 and Sector 2. (c-d) The $I^{1/2}$ - q^2 curves plotted following the Debye-Bueche model, $(I(q))^{-1/2} = K(a^3Q)^{-1/2}(1 + a^2q^2)$, where K is a constant, Q is the scattering invariant, and a is the correlation length, which described the average domain size. The correlation lengths were fitted from the linear region at low- q range (0.0012 - 0.003 \AA^{-1}). (e) Representative fit region (0.001 - 0.003 \AA^{-1}) of the $I^{1/2}$ - q^2 curves. (f) Representative linear fits of the $I^{1/2}$ - q^2 curves, where the resultant correlation lengths

equal to the values of $(\text{slope}/\text{intercept})^{-1/2}$ [\AA].

Fig. S43. Correlation length as a function of film strain of the P(NDI2OD-T2) thin film. The correlation length in both section 1 and section 2 first decreased by ~ 50 nm during the initial strain ($<5\%$), likely due to transverse compression, and subsequently increased from ~ 150 nm to ~ 185 nm as strain progressed. This trend complements the micro-scale aggregation observed by AFM (Fig. S41).

6. *The UV-Vis spectrum of the blend film is unclear. The spectral features that appear in the blend film is significantly shifted compared to neat P(NDI2OD-T2). It is unclear whether this shift originates from the P(NDI2OD-T2) component or from PO10. The authors should include a strain-dependent UV-Vis spectrum of neat PO10 films to determine the individual contributions of each material in the blend. This would allow them to confidently attribute the observed blue shift in the blend to a specific molecular origin.*

Our response: Thanks for the comment. To clarify the origin of the absorption shift in the blend film under strain, we measured the strain-dependent UV-vis absorption spectra of neat PO10 films. The result indicates that PO10 exhibits a main absorption peak at 578 nm within 400-700 nm, which undergoes only a slight blue shift and narrowing upon stretching. In contrast, the blue shift observed in the blend films

primarily occurs in the 720-800 nm region, consistent with the absorption shift of neat P(NDI2OD-T2). These results confirm that the strain-induced spectral shift in the blends arises from the P(NDI2OD-T2) component. To clarify this point, we revised the following sentence in the manuscript.

Main text (Page 14, line 26):

“Similar to the neat thin film, the P(NDI2OD-T2) component in the blend showed blue-shifted absorption, during stretch from 0% to 30% strain, while PO10 contributes only minor changes within its own absorption range (Fig. 4e and Fig. S52).”

Fig. S52. Normalized absorption spectra of the neat PO10 thin film under different stretch strains with a film-on-elastomer sample.

7. While the manuscript connects morphological evolution to device performance, the authors could elaborate further on how the relatively subtle strain-dependent optical changes (only a few nm shifts in absorption peaks) influence charge generation, transport, or recombination in OPV devices.

Our response: Thanks for the comment. Although the strain-induced absorption changes are relatively small (~10 nm peak shift and a decreased shoulder intensity at ~790 nm), they signify meaningful variations in electronic coupling and molecular packing order. These arise from chain conformational adjustment, orientation, and partial crystallite disruption, which collectively reshape the morphology and thus

influence charge generation, transport, and recombination in the devices. To clarify these correlations, we performed measurements of photocurrent density-effective voltage ($J_{\text{ph}}-V_{\text{eff}}$) curves, photo-CELIV, and light intensity dependent $J-V$ curves for stretchable OPV devices under different strains (0%, 10%, 20%, and 30%).

From the $J_{\text{ph}}-V_{\text{eff}}$ analysis, the exciton dissociation and charge collection efficiency, $P(E, T)$, decreased slightly from 98.86% for pre-stretched condition to 97.93%, 96.85%, and 95.53% at 10%, 20%, and 30% strain, respectively, indicating reduced charge generation under strain. The photo-CELIV results further revealed a gradual decline in charge mobility from 6.59×10^{-5} to $4.13 \times 10^{-5} \text{ cm}^2 \text{ V}^{-1} \text{ s}^{-1}$ as the strain increased from 0% to 30%. Moreover, the light intensity (P_{light}) dependence of V_{OC} can be described by $V_{\text{OC}} \propto n \frac{KT}{q} \ln(P_{\text{light}})$, yielding n values of 1.25, 1.42, 1.46, and 1.58 for devices strained at 0%, 10%, 20%, and 30%, respectively, suggesting progressively enhanced trap-assisted recombination. Coupled with the AFM-IR images, the results demonstrate that even the subtle strain-dependent optical shifts, originating from local order-disorder transitions to morphological aggregation, translate into measurable deterioration in charge generation and transport, along with increased recombination losses. To clarify this, we have revised the following section in the revised manuscript.

Main text (Page 15, line 1):

“Such decline was primarily due to the decrease in both J_{SC} and FF (Fig. 4f), which correlate with charge generation, transport, and recombination in the devices.

The exciton dissociation and charge collection efficiency, $P(E, T)$, extracted from photocurrent density-effective voltage ($J_{\text{ph}}-V_{\text{eff}}$) curves, decreased from 98.86% for pre-stretched condition to 97.93%, 96.85%, and 95.53% at 10%, 20%, and 30% strain, respectively, indicating reduced charge generation (Fig. S54). Photo-CELIV analysis revealed a concurrent decline in charge mobility from 6.59×10^{-5} to $4.13 \times 10^{-5} \text{ cm}^2 \text{ V}^{-1} \text{ s}^{-1}$ (Fig. S55). The light intensity (P_{light}) dependence of V_{OC} , described by $V_{\text{OC}} \propto n \frac{KT}{q} \ln(P_{\text{light}})$, yields n values of 1.25, 1.42, 1.46, and 1.58 as strain increased, signifying enhanced trap-assisted recombination (Fig. S56). To further elucidate this, morphological changes upon thin film stretched by nano-infrared spectroscopy atomic

force microscopy (AFM-IR). As shown in Fig. 4g-i, P(NDI2OD-T2) domains probed with 843 cm^{-1} exhibit nanofibrillar network morphology in the as-cast blend film (Fig. S56), which is retained when 10% strain was applied, but transformed into large aggregates at 30% strain. This morphological transformation arises from strain-induced chain orientation, intrachain torsion, and crystalline destruction, leading to local order-disorder transitions. These combined absorption, electrical, and morphological results demonstrate that strain-induced structural evolution directly deteriorates charge generation and transport while promoting recombination losses, ultimately accounting for the observed performance degradation. To assess the effect of cyclic strain, PO10:P(NDI2OD-T2) films on PDMS were stretched at 20% for 300 cycles. AFM images show that the fibrillar morphology gradually blurred with repeated cycling (Fig. S58), causing some decline in device performance; however, the device retained over 80% of its initial PCE (Fig. S59 and Table S4). These results indicate that the overall nanostructure and device performance remain largely preserved under moderate morphological relaxation.”

Supplementary Information:

Fig. S54. $J_{\text{ph}}-V_{\text{eff}}$ curves of PO10:P(NDI2OD-T2) based stretchable OPV device under different strains.

Fig. S55. Photo-CELIV curves and the corresponding mobilities (μ) of PO10:P(NDI2OD-T2) based stretchable OPV device under different strains.

Fig. S56. Light-intensity (P_{light}) dependency of V_{oc} of PO10:P(NDI2OD-T2) based stretchable OPV device under different strains.

8. The authors could expand the discussion of prior works on strain-induced morphology changes in conjugated polymers. For example, Wu *et al.* (<https://www.nature.com/articles/s41467-023-44099-w>) introduced the “relative

stretchability” (rS, the ratio of DR and rDoC) to quantitatively analyze the strain induced morphological evolution and crystal breakage. Comments on whether your system follows similar entropic (chain alignment, from DR) and enthalpic (crystallite amorphization/breakage, from rDoC) dissipation pathways would be helpful for the readers. In addition, Zheng et al. (<https://pubs.acs.org/doi/full/10.1021/jacs.2c00072>) summarized the molecular and morphological processes expected under strain, for example, chain alignment and strain-induced backbone planarization through monomer torsional rotation and then outlined how such processes map onto mobility retention. Since this manuscript shows direct evidence on monomer rotation to planarize the chain under strain, highlighting how it supports or refines the assumptions in this review paper would be helpful.

Our response: Thanks for the comment. We have now expanded the discussion of prior works to better contextualize our findings. Wu et al. (Nat. Commun. 2023, 14, 8382) proposed the concept of relative stretchability (rS), which integrates the effects of chain alignment (DR) and crystallite breakage (rDoC) to describe dual entropic and enthalpic dissipation pathways during strain. In our system, both effects were directly captured, where chain alignment and crystallite destruction were evidenced by NEXAFS and TReXS/GIWAXS, respectively. This observation aligns with the dual-dissipation framework of Wu et al. but further refines it by revealing a strain-dependent temporal evolution. In relation to the review by Zheng et al. (J. Am. Chem. Soc. 2022, 144, 4699), we note that their discussion of strain-induced backbone behavior focused on orientation and rotation. Our in situ and MD simulation data complement these findings: we observe increased dihedral torsion angles under strain, implying enhanced rotational freedom. These results clarify that strain-induced molecular responses are strongly dependent on side-chain architecture and packing constraints. To clarify this, we added the following points in the Discussion section of the manuscript.

Main text (Page 15, line 28):

“Recent studies have identified entropic chain alignment and enthalpic crystallite breakage as multimodal energy dissipation pathways for mechanical robustness [55].

Both effects are directly observed in our system, with a strain-dependent temporal evolution further revealed [56]. Consistent with a recent review highlighting the role of chain alignment and torsional rotation in mechanical-electronic coupling, the in situ and simulation data show increased torsional freedom under strain, likely governed by backbone architecture.”

References:

55 Wu, H.-C. et al. Highly stretchable polymer semiconductor thin films with multi-modal energy dissipation and high relative stretchability. *Nat. Comm.* **14**, 8382 (2023).

56 Zheng, Y. et al. A molecular design approach towards elastic and multifunctional polymer electronics. *Nat. Comm.* **12**, 5701 (2021).

Reviewer #2 (Remarks to the Author):

The manuscript presents a fundamental investigation into the structural evolution of a conjugated polymer under tensile stress, utilizing a combination of NEXAFS, TReXS, and RSoXS techniques. The studied material is further demonstrated in a fully stretchable photovoltaic device. Overall, the manuscript is well written, and the comprehensive suite of characterizations are clearly correlated to provide a coherent mesoscale picture that advances the fundamental understanding of intrinsically stretchable organic semiconductors during dynamic mechanical deformation. I recommend this work for publication, provided that the following concerns and questions are satisfactorily addressed:

Q1: In comparing the simulated NEXAFS spectra of P(NDI2OD-T2) with the experimental results in Figure 1c, there remain noticeable discrepancies, particularly around 285.1 eV. What might be the origin of these mismatches? Additionally, the major and minor tick marks on the x-axis of Figure 1c are not clearly visible and should be improved for clarity.

Our response: Thanks for the comment. The slight mismatch in peak positions between the simulated and experimental spectra (e.g., 285.0 eV in experiment vs. 284.8 eV in simulation) mainly arises from intrinsic limitations of first-principles NEXAFS calculations. In our case, the simulations were based on the simplified BT-NDI structural unit rather than the full P(NDI2OD-T2) polymer, so long-range conjugation and solid-state effects are not fully captured. Moreover, the PBE-GGA functional is known to underestimate bandgaps, which compresses the spectra along the energy axis even after applying a dilation factor, while the lack of environmental interactions such as interchain packing and substrate effects in the vacuum-based model can also shift resonance energies. Finally, the applied alignment (284.5 eV for the C K-edge) and Gaussian broadening inevitably introduce small offsets. Overall, the deviation of ~0.2-0.3 eV is within the typical range reported for DFT-based NEXAFS simulations and does not affect the spectral assignments or conclusions of NEXAFS analysis. To clarify this, we added the following sentences in the Method section.

Method:

“We note that despite these corrections, small residual discrepancy of $\sim 0.2\text{-}0.3$ eV remains, as the simplified molecular model cannot fully capture the long-range conjugation, and the vacuum-based calculations cannot account for solid-state interactions. These deviations are commonly accepted in DFT-based NEXAFS simulations and do not affect the reliability of spectral assignments [34,37,62]”

We also revised the Figure 1c to clearly show the major and minor ticks as follows.

Fig. 1. (c) Experimental (TFY) and simulated NEXAFS spectra of P(NDI2OD-T2) in the C $1s \rightarrow \pi^*$ transitions range (283.6-286.0 eV), and the contributions from the NDI and BT units.

Q2: Regarding the free-standing experimental setup, as the majority of the film remains supported by the PDMS substrate, could this result in differences between the local strain experienced near the PDMS opening and the overall applied strain? How is the strain distributed across the film, and is there any quantification or mapping of potential strain gradients?

Our response: Thanks for the comment. Finite element analysis (FEA) of the PDMS-supported freestanding geometry (a 4 mm diameter circular hole at the center of the PDMS) show that, under a perfect-bonding assumption, the unsupported thin film can experience high local strain amplification near the hole edge: for a 30% applied overall

strain of the sample, the model predicts a local strain of 79% at the edge, due to stress concentration. The simulation also shows that the central region of the film exhibits uniaxial, spatially uniform in-plane strain, which is over an area much larger than the X-ray beam footprint, confirming that the RSoXS/TReXS measurements probe a uniaxial strain state.

In the FEA, no slip between the film and PDMS was assumed; however, experimental observations indicate that no macroscopic bending occurs in practice, suggesting that strain amplification is partially relaxed and the actual film strain is closer to the nominal applied strain. This setup therefore provides the most practical approach for realizing uniaxial stretching of thin polymer films during in situ X-ray scattering experiments. We report the nominal applied strain in the manuscript while noting this limitation and include the FEA maps in the Supplementary Information for transparency.

Main text (Page 7, line 17):

“The PDMS-supported freestanding configuration provides a practical approach to impose uniaxial strain on thin polymer films during in situ X-ray scattering experiments, as validated by simulated strain and stress maps (Fig. S17 and S18).”

Supplementary Information:

Fig. S17. Finite element analysis (FEA): (a) Simulated models of the P(NDI2OD-T2)

thin film on a PDMS substrate with a central circular opening. (b) Simulated strain map, showing that the overall strain applied to the PDMS substrate (30%) leads to local strain amplification (79%) near the hole edge. The PDMS substrate and central hole have initial lengths of 15 cm and 0.4 cm, respectively. The inset numbers are in units of μm . (c) Side-view strain map, showing significant macroscopic bending under the perfect bonding assumption between the PDMS and P(NDI2OD-T2) film. (d) Simulated stress map, showing the stress concentration at the central hole, where the central 0.2 mm region of the unsupported film exhibits a nearly uniform uniaxial strain.

Notes: FEA simulations of the samples for TReXS and RSoXS were performed using Abaqus. The PDMS substrate was modeled as a rectangular block with dimensions of $10\text{ cm} \times 15\text{ cm} \times 0.5\text{ cm}$, containing a circular opening of 4 mm in diameter at the center. A thin P(NDI2OD-T2) film (thickness: $0.2\ \mu\text{m}$) was conformally placed on the top surface of the PDMS, fully covering the hole region. A perfect bonding condition (no slip) was assumed between the P(NDI2OD-T2) film and PDMS, and the two layers were meshed using shared nodes. To simplify the computation while retaining the strain-distribution trend, both materials were described by linear elastic constitutive relations. The elastic moduli were set as 1.74 MPa for PDMS and 63.72 MPa for P(NDI2OD-T2). A static step analysis was used, where one side of the model was fully fixed while a 30% tensile strain was applied along the x-axis to the opposite side.

Fig. S18. The photos showing a P(NDI2OD-T2) thin film with 0% and 30% strain on the tensile tester. Such photos indicate no macroscopic bending occurs in practice, which reduces strain amplification and brings the actual film strain closer to the applied nominal value.

Q3: For the TReXS and RSoXS experimental setups, which employ a transmission geometry, how is it ensured that the X-ray beam does not also penetrate and contribute signal from the PDMS substrate? What is the beam size relative to the size of the unsupported region (hole)? Additionally, since the scattering intensity is correlated with film thickness—and considering that the film thickness is likely reduced during stretching—is this variation reflected or corrected for in the analysis (for example, in Figure 2d)?

Our response: Thanks for the comment. The PDMS substrate contains a 4 mm diameter hole, which is substantially larger than the incident X-ray beam size ($20 \times 200 \mu\text{m}^2$ for TReXS and $200 \times 200 \mu\text{m}^2$ for RSoXS), ensuring that the beam fully passes through the unsupported film region. In addition, no diffraction contribution from PDMS is observed: PDMS typically shows a reflection near $q \sim 0.40 \text{ \AA}^{-1}$, whereas our TReXS data remain flat in this range, confirming that the measured scattering originates solely from the active layer.

For RSoXS measurements, the PDMS substrate thickness (0.5-0.8 mm) effectively blocks the C K-edge soft X-rays, preventing beam penetration (hundreds of nanometers for attenuation length at $\sim 285 \text{ eV}$) and background scattering from the substrate. Therefore, the signals collected in both TReXS and RSoXS arise exclusively from the polymer thin films.

Regarding the scattering intensity in TReXS analysis, we applied normalization to account for possible variations in film thickness upon stretching. Specifically, the scattering intensity was normalized to the intensity at q position of 0.37 \AA^{-1} under 0% strain, ensuring consistent comparison across all strain levels. To clarify this, we added the following notes in the Method section.

Method:

TReXS

“The P(NDI2OD-T2) thin films were transferred onto PDMS substrates with a central circular hole (4 mm in diameter), as illustrated in Fig. S3b. The X-ray beam size was $20 \mu\text{m} \times 200 \mu\text{m}$, ensuring that the beam fully passed through the unsupported film

region.”

“To ensure consistent comparison across all strain levels, the scattering intensity was normalized to the value at $q = 0.37 \text{ \AA}^{-1}$ at 0% strain. No diffraction contribution from PDMS was observed; PDMS typically shows a reflection at $q \sim 0.40 \text{ \AA}^{-1}$ (Fig. S61), whereas our TRexS data remain flat in this range, confirming that the measured scattering originates solely from the polymer thin film.”

NEXAFS and RSoXS

“The sample preparation followed the same procedure as shown in Fig. S3b. The beam size is $200 \mu\text{m} \times 200 \mu\text{m}$, ensuring that the beam fully passed through the unsupported film region.”

Fig. S61. WAXS (a) 2D image and (b) I - q curves averaged in the parallel (para.) and perpendicular (perp.) directions.

Q4: While optical microscopy determined the crack-onset of P(NDI2OD-T2) to occur at strains above 50% (Fig. S2), the macroscopic stress-strain curve exhibits mechanical failure before reaching 35% strain (Fig. 4a). Can the authors clarify the origin of this apparent discrepancy?

Our response: Thanks for the comment. The apparent discrepancy arises because the two measurements probe different mechanical conditions and failure criteria. The optical microscopy test characterizes the crack-onset strain (COS) of a thin film supported on a compliant PDMS substrate, where strain is partially relaxed through the substrate, and visible surface cracks appear only after local stress concentration has

accumulated. In contrast, the macroscopic stress-strain curve in Fig. 4a was measured from a free-standing film, where mechanical failure is defined by a sudden stress drop due to complete fracture. Therefore, the supported film can appear to sustain higher nominal strain before visible cracking, even though local strain in the film may already be high. Such differences between supported and free-standing geometries have also been reported in stretchable semiconducting thin films. To clarify this distinction, we added the following sentence in the manuscript.

Main text (Page 13, line 23):

“Note that the crack-onset strain from optical microscopy (supported film) is not directly comparable to the fracture strain from tensile testing (free-standing film), as the former reflects delayed surface cracking under strain relaxation by the substrate, whereas the latter represents complete mechanical failure of the film.”

Q5: The observation of a two-stage strain response corresponding to distinct stress dissipation mechanisms in P(NDI2OD-T2) is intriguing. To what extent is this behavior generalizable to other conjugated polymers, such as PO10, already studied in this work?

Our response: Thanks for the comment. The observed two-stage strain response in P(NDI2OD-T2) is closely related to its mechanical behavior, including yielding and hardening, as reflected in the stress-strain curves. This behavior arises from how stress is dissipated within conjugated polymers, which varies with molecular structure and solid-state morphology. In this work, P(NDI2OD-T2) is a highly crystalline polymer, whereas PO10 exhibits much lower crystallinity. Therefore, it is reasonable to expect distinct strain-response characteristics for PO10. Meanwhile, to observe a similar two-stage morphological evolution using in situ RSoXS, it is necessary to identify an appropriate soft X-ray energy that can enhance the scattering contrast, particularly the orientational contrast. However, such an energy is not always experimentally accessible. The use of P(NDI2OD-T2) here thus serves as an example to demonstrate how a combination of X-ray scattering and spectroscopy techniques can reveal multi-scale structural evolution in conjugated thin films and its correlation with electronic

performance.

Indeed, we are currently extending this study to polymers with lower or near-amorphous crystallinity using in situ RSoXS and molecular dynamics simulations for stretchable organic solar cells and organic light-emitting diodes. Preliminary results indicate that strain-induced orientation occurs across molecular to mesoscale structures, accompanied by torsional conformational changes, which are consistent with the observed in this work. We are continuing this analysis and plan to report these findings in a future publication. To clarify this point, we have revised the following sentences in the revised manuscript.

Main text (Page 17, line 11):

“Although the two-step morphological response is demonstrated using the highly crystalline P(NDI2OD-T2), its occurrence is expected to depend on the molecular structure and thin-film morphology. Ongoing studies on polymers with lower crystallinity or near-amorphous states aim to extend these insights and examine the generality of the observed behavior. Nevertheless, our findings establish a mechanistic framework for understanding how conjugated polymer systems maintain device functionality under large deformations.”

Q6: The demonstration of robust photovoltaic device performance under up to 50% applied strain is impressive. Is it possible to directly characterize the charge carrier mobility of P(NDI2OD-T2) under strain? Such measurements would provide important insight into the relationship between mechanical deformation, structural evolution, and charge transport properties.

Our response: Thanks for the comment. To further elucidate the relationship between mechanical deformation and charge transport, we have evaluated the charge carrier mobility of the stretchable OPV device under strain using photo-CELIV measurements. The mobility decreased from 6.59×10^{-5} at 0% strain to $4.13 \times 10^{-5} \text{ cm}^2 \text{ V}^{-1} \text{ s}^{-1}$ at 30% strain. We have also evaluated the charge generation and recombination by photocurrent density-effective voltage ($J_{\text{ph}}-V_{\text{eff}}$) curves and light intensity dependence of V_{OC} ,

respectively. Coupled with the AFM-IR images, the results collectively indicate that the strain-induced absorption shifts and structural evolution, from local order-disorder transitions to morphological aggregation, translate into measurable deterioration in charge generation and transport, accompanied by enhanced recombination losses. To clarify this, we revised the following section in the manuscript.

Main text (Page 15, line 3):

“The exciton dissociation and charge collection efficiency, $P(E,T)$, extracted from photocurrent density-effective voltage ($J_{\text{ph}}-V_{\text{eff}}$) curves, decreased from 98.86% for pre-stretched condition to 97.93%, 96.85%, and 95.53% at 10%, 20%, and 30% strain, respectively, indicating reduced charge generation (Fig. S54). Photo-CELIV analysis revealed a concurrent decline in charge mobility from 6.59×10^{-5} to 4.13×10^{-5} $\text{cm}^2 \text{V}^{-1} \text{s}^{-1}$ (Fig. S55). The light intensity (P_{light}) dependence of V_{OC} , described by $V_{\text{OC}} \propto n \frac{KT}{q} \ln(P_{\text{light}})$, yields n values of 1.25, 1.42, 1.46, and 1.58 as strain increased, signifying enhanced trap-assisted recombination (Fig. S56). To further elucidate this, morphological changes upon thin film stretched by nano-infrared spectroscopy atomic force microscopy (AFM-IR). As shown in Fig. 4g-i, P(NDI2OD-T2) domains probed with 843 cm^{-1} exhibit nanofibrillar network morphology in the as-cast blend film (Fig. S57), which is retained when 10% strain was applied, but transformed into large aggregates at 30% strain. This morphological transformation arises from strain-induced chain orientation, intrachain torsion, and crystalline destruction, leading to local order-disorder transitions. These combined absorption, electrical, and morphological results demonstrate that strain-induced structural evolution directly deteriorates charge generation and transport while promoting recombination losses, ultimately accounting for the observed performance degradation.”

Supplementary Information:

Fig. S54. J_{ph} - V_{eff} curves of PO10:P(NDI2OD-T2) based stretchable OPV device under different strains.

Fig. S55. Photo-CELIV curves and the corresponding mobilities (μ) of PO10:P(NDI2OD-T2) based stretchable OPV device under different strains.

Fig. S56. Light-intensity (P_{light}) dependency of V_{OC} of PO10:P(NDI2OD-T2) based stretchable OPV device under different strains.

Q7: For real-world applications, stretchable semiconductors will undergo repeated cyclic stretching rather than a single extension. How do the authors anticipate that cyclic mechanical fatigue will influence the nanostructure and electrical performance over time? Are there any preliminary results or plans to study such effects?

Our response: Thanks for the comment. To assess the influence of cyclic mechanical deformation on nanostructure and device stability, PO10:P(NDI2OD-T2) blend films were transferred onto PDMS substrates and subjected to cyclic stretching at 20% strain for 100 and 300 cycles. The initially well-defined fibrillar morphology became progressively blurred with increased surface roughness after repeated stretching, indicating partial disruption of the nanofibrillar network. Correspondingly, the stretchable OPV devices exhibited a gradual decline in performance but retained over 80% of its initial PCE after 300 strain cycles, demonstrating good mechanical durability. These results suggest that while cyclic strain introduces moderate morphological relaxation and roughening, the overall nanostructure integrity and device function remain largely preserved. Further studies are planned to systematically probe the fatigue-induced morphological evolution and charge transport behavior under more

prolonged cyclic deformation. To clarify this, we added the following sentences in the manuscript.

Main text (Page 15, line 19):

“To assess the effect of cyclic strain, PO10:P(NDI2OD-T2) films on PDMS were stretched at 20% for 300 cycles. AFM images show that the fibrillar morphology gradually blurred with repeated cycling (Fig. S58), causing some decline in device performance; however, the device retained over 80% of its initial PCE (Fig. S59 and Table S4). These results indicate that the overall nanostructure and device performance remain largely preserved under moderate morphological relaxation.”

Supplementary Information:

Fig. S58. AFM height images of PO10:P(NDI2OD-T2) thin films on PDMS substrates under (a) pre-stretched condition, (b) after 100 cycles at 20% strain, and (c) after 300 cycles at 20% strain.

Fig. S59. (a) $J-V$ curves and (b) normalized PCE of stretchable OPV device based on PO10:P(NDI2OD-T2) thin films under pre-stretched condition, after 100 cycles at 20%

strain, and after 300 cycles at 20% strain.

Table S4. Device parameters of stretchable OPV device based on PO10:P(NDI2OD-T2) thin films under cyclic stretch at 20% strain.

Cycles	V_{OC} [V]	J_{SC} [mA cm ⁻²]	FF [%]	PCE [%]	Normalized PCE
0	0.855	13.91	55.67	6.62	1
100	0.813	14.47	51.73	6.09	0.92
300	0.797	13.94	50.50	5.61	0.85

Q8: The molecular weight of the P(NDI2OD-T2) is very high, 131k; providing the P(NDI2OD-T2) GPC traces and the calibration curves will be helpful. Besides, material information (synthetic procedure, molecular weight) on PO10 and PNDIT-F3N is missing.

Our response: We thank the reviewer for this valuable suggestion. We have included the GPC molecular-weight distribution of P(NDI2OD-T2) in the Supplementary Information (Fig. S60). The key GPC parameters are provided (Retention time = 24.287 min; M_n = 131 kDa; M_w = 178 kDa; Dispersity = 1.356). Molecular weights are reported as polystyrene (PS) equivalents. The GPC experimental details in the Method section have also been revised.

Fig. S60. Molecular weight distribution and cumulative curves of P(NDI2OD-T2).

The polymer PO10 was synthesized according to our previous report with number averaged molecular weight of 24.4 kDa and dispersity of 2.15 [Macromolecules 56, 8928-8938 (2023)]. The number-average molecular weight of PNDIT-F3N was determined as 12.3 kDa by a Waters GPC 2410 with tetrahydrofuran (THF) as the eluent using linear polystyrene standards. Such key information has been added in the Method section of the manuscript as follows.

Method:

“P(NDI2OD-T2) was synthesized according to a reported procedure [61]. Its molecular weight was characterized by high-temperature gel permeation chromatography (GPC) on a Tosoh EcoSEC HT system equipped with three TSKgel GMHhr-H(S) HT columns in series, using 1,2,4-trichlorobenzene as the eluent at 135 °C. Calibration was carried out against narrow-dispersity (\mathcal{D}) polystyrene standards. The number-average molecular weight (M_n) and dispersity of P(NDI2OD-T2) were determined to be 131 kDa and 1.356, respectively. The corresponding molecular-weight distribution curve is shown in Fig. S36. The polymer PO10 was synthesized as described in our previous work [52], with $M_n = 24.4$ kDa and $\mathcal{D} = 2.15$. The M_n and \mathcal{D} of PNDIT-F3N were determined to be 6.25 kDa and 1.34 by GPC (Waters 2410) using tetrahydrofuran (THF) as the eluent and linear polystyrene standards for calibration.”

[52] Li, K. et al. 8.0% Efficient all-polymer solar cells based on novel starburst polymer acceptors. *Sci. China Chem.* **61**, 576-583 (2018).

[61] Luo, X. et al. Intrinsically Stretchable Organic Photovoltaic Thin Films Enabled by Optimized Donor–Acceptor Pairing. *Macromolecules* **56**, 8928-8938 (2023).

Reviewer #3 (Remarks to the Author):

This manuscript presents a novel and comprehensive approach to study the multiscale structural evolution of conjugated polymer thin films under strain, using a powerful combination of advanced in situ X-ray techniques (NEXAFS, TReXS, RSoXS). The study provides important mechanistic insights into stress-dissipation pathways, including chain orientation changes, intrachain torsion, crystallite disruption (via “peeling” and “slippage” mechanisms), and a “two-step” morphological reorganization. Such structural changes are successfully connected to mechanical and optoelectronic properties. The experimental design is rigorous, the correlative multimodal methodology is state-of-the-art, and the results offer valuable guidance for the rational design of durable stretchable electronics. Given the novelty of the characterization method and the depth of the mechanistic insights, I strongly recommend publication after minor revisions.

(1) While the title emphasizes the study of structural evolution on organic photovoltaic, the Introduction does not clearly articulate how key morphological changes under strain impact photovoltaic metrics. A more explicit discussion connecting the multiscale structural changes to performance-limiting mechanisms in organic solar cells would enhance the impact of the work.

Our response: Thanks for the comment. In the revised manuscript, we have expanded the Introduction to explicitly connect strain-induced morphological changes with performance-limiting mechanisms in organic solar cells. In particular, we now highlight how chain alignment and entanglement dissipate stress and how ductile polymer networks can shield brittle acceptor crystallites, thereby mitigating efficiency losses under high strain. To provide broader context, we also referenced reported mechanisms of strain-induced structural evolution in stretchable OFETs and OLEDs, illustrating how multiscale morphological changes affect charge transport and optoelectronic properties across different device types. These additions aim to present a more comprehensive picture of structure-property relationships under mechanical deformation. The following sentences were added to the Introduction section:

Main text: (Page 3, line 19):

“Nevertheless, conjugated polymers have enabled intrinsically stretchable electronics by molecular design and microstructural control. In organic field-effect transistors (OFETs), high-molecular-weight polymers shows retained crystalline domains and reversible chain alignment under >100% strain, thereby preserving electrical performance [19]. Random terpolymer design can suppress crystallinity and long-range order, while maintaining short-range ordered aggregates, resulting in high mobility with improved stretchability [20]. In organic photovoltaics (OPVs), polymer-polymer blends outperform polymer-small molecule blends because gradual chain alignment and entangled networks dissipate stress while sustaining charge transport pathways [21]. The introduction of high-molecular-weight conjugated polymers offers soft, amorphous regions with entangled and tie chains for stress dissipation, shielding the brittle acceptor crystallites and mitigating efficiency loss [22]. In organic light-emitting diodes (OLEDs), plasticizer incorporation transforms brittle conjugated polymers into fibrillar morphologies, resolving the conductivity-stretchability trade-off [23]. Despite these advances, direct in situ probing of the multiscale structural changes in conjugated polymer thin films, and how they collectively govern stress dissipation from the molecular to the mesoscale remains an outstanding challenge.”

References:

19. Wu, H.-C. et al. Highly stretchable polymer semiconductor thin films with multi-modal energy dissipation and high relative stretchability. *Nat. Comm.* **14**, 8382 (2023).
20. Mun, J. et al. A design strategy for high mobility stretchable polymer semiconductors. *Nat. Comm.* **12**, 3572 (2021).
21. Wang, Z. et al. Revealing the Strain-Induced Morphological, Mechanical, and Photovoltaic Evolution of Self-Encapsulated and Semitransparent Intrinsically Stretchable Organic Solar Cells. *ACS Mater. Lett.* **6**, 1811-1819 (2024).
22. Peng, Z. et al. Unraveling the Stretch-Induced Microstructural Evolution and Morphology–Stretchability Relationships of High-Performance Ternary Organic Photovoltaic Blends. *Adv. Mater.* **35**, 2207884 (2023).

23. Jin-Hoon Kim, Jin-Woo Park, Intrinsically stretchable organic light-emitting diodes. *Sci. Adv.* **2021**, 7, eabd9715.

(2) The angle-resolved NEXAFS shows increased α -angles of polymer chains under strain. Are similar changes observed within the crystalline domains? Since face-on π - π stacking is crucial for vertical charge transport in OSCs, clarification on whether the crystallite orientation is similarly affected would be helpful.

Our response: Thanks for the comment. The increased α -angles observed in angle-resolved NEXAFS reflect an overall enhancement of face-on chain orientation, including both crystalline and amorphous regions. To explicitly probe crystalline domains, we performed GIWAXS measurements on strained films transferred to silicon wafers. The azimuthal intensity distribution of the π - π stacking reflection showed narrowing when probed perpendicular to the strain direction and broadening when probed parallel. This indicates enhanced face-on π - π stacking orientation using X-ray to probe the perpendicular direction, consistent with the NEXAFS results. Moreover, the contrasting changes in pole figure width suggest reduced twisting of crystallites about the (001) axis but increased tilting about the (100) axis. Together, these findings demonstrate that crystallite deformation and orientation contribute to irreversible structural adaptation for stress dissipation under strain. To clarify this point, we have added the following sentences in the manuscript.

Main Text (Insertion, Page 9, Line 7):

“To study the π - π stacking developments during stretch, the films were stretched to the desired strains and then placed on silicon wafers for grazing incidence wide angle X-ray scattering (GIWAXS) measurements at the beam energy of 10 keV. This measurement was performed with orthogonal X-ray directions including parallel and perpendicular conditions (Fig. S28). The 0% strain thin film shows nearly identical reflections (Fig. S29), which support that $\beta_x \approx \beta_y$ before stretching as demonstrated by NEXAFS. Meanwhile, the (100) peak area in both q_x and q_y decreases rapidly at small strains (10-20%) and then more gradually at higher strains, with a stronger reduction

along q_x (Fig. S30-32, Table S1-S2). This behavior is consistent with the observations at 2476.26 eV in TRexS, indicating that the resonant excitation is not sensitive to the crystalline deformation. The azimuthal intensity of the π - π stacking reflection was used to evaluate crystallite orientation relative to the substrate (Fig. S33). With increasing strain, the pole figures show narrowing for the perpendicular condition and broadening for the parallel condition. Therefore, an enhanced face-on orientation of π - π stacking is observed when X-rays probe perpendicular to the SD, in agreement with the angle-resolved NEXAFS results. The opposite trend in pole figure widths suggests reduced twisting of crystallites about the (001) axis but increased tilting about the (100) axis, consistent with the tilted crystallite orientation observed in flow-induced alignment [43]. Together, these combined mechanisms of crystallite deformation and orientation provide irreversible structural adaptation pathways for stress dissipation.”

Reference:

[43] Persson, N.E., Engmann, S., Richter, L.J. & DeLongchamp, D.M. In Situ Observation of Alignment Templating by Seed Crystals in Highly Anisotropic Polymer Transistors. *Chem. Mater.* **31**, 4133-4147 (2019).

Supplementary Information:

Fig. S28. (a) Schematic representation of GIWAXS geometries for strained P(NDI2OD-T2) thin film, where the scattering images are collected with the incident X-ray roughly perpendicular (perp.) or parallel (para.) to the SD. (b) GIWAXS patterns of P(NDI2OD-T2) thin films with various strains measured with the incident X-ray beam (b) perpendicular (q_x) and (c) parallel (q_y) to the SD.

Fig. S33. (a) Illustration of the circular average for the generation of (010) pole figures and (b) fitting example for the pole figures using Gaussian function. (c) (010) pole figure full width at half maximum (FWHM) as functions of strain.

(3) The observed decrease in crystallinity during stretching is reasonable due to the rigid backbone of conjugated polymers. However, it remains unclear why the scattering intensity continues to decrease upon strain release.

Our response: Thanks for the comment. The continued decrease in scattering intensity during strain release can be rationalized as follows: (i) in the initial stage of release (30% \rightarrow ~20% strain), residual stress remains above the critical threshold for crystal stability, causing further disruption of crystallites even as the macroscopic strain decreases; (ii) below ~20% strain, the applied stress falls below this threshold, so no additional crystal destruction occurs, but the previously disrupted structures cannot self-reorganize due to irreversible chain slip and peeling. As a result, the scattering intensity does not

recover upon full release. To clarify this, we added the following sentences in the manuscript.

Main text (Page 8, line 18):

“Upon releasing the film, the scattering intensities continued to decrease between 30% and 20% strain, likely due to residual stress remaining above the critical threshold for crystallite destruction. Below ~20% strain, no further decrease was observed; however, neither reflection recovered, indicating a permanent loss of crystallinity for both directions.”

(4) The use of $\Delta\phi$ to quantify morphological orientation is well-executed and convincingly correlates with mechanical strain (σ - ϵ). However, the physical origin of the “X”-shaped contrast in the RSoXS images should be clarified due to the structural complexity of the strained thin film. Meanwhile, how are the ϕ positions determined?

Our response: Thanks for the comment. The “X”-shaped contrast in the RSoXS patterns originates from oriented fibrillar structures, as supported by the consistent “X”-like features observed in the FFT analysis of AFM images. We acknowledge that the morphology of strained thin films is complex, where domain and orientational contrasts are intertwined, making it challenging to decouple their individual contributions. Nevertheless, orientational contrast, which can be quantitatively analyzed, provides a reasonable approximation, as the collective alignment of polymer chains forms oriented fibrils. Although this approach may not fully capture the complete complexity of the system, it effectively accounts for the main source of contrast. In particular, we observed a pronounced increase in orientational contrast in the xy- and xz-planes at 284.0 eV under strain, consistent with the experimental ISI results. To further clarify this point, we have added in situ RSoXS images of the P(NDI2OD-T2) thin film at various strains during stretching and releasing, which show that the “X”-shaped feature is absent when probed at 285.0 eV. These experimental observations and calculations strongly support that the “X”-shaped scattering feature originates from the strain-induced orientation of polymer chains within the fibrillar morphology, with its visibility

being highly sensitive to the selected X-ray energy. To address this, we added the following sentence in the manuscript.

Main text (Page 10, line 27):

“We note that the strained thin-film morphology is complex, as domain and orientational contrasts are intertwined and difficult to decouple. Nevertheless, the consistent “X” shape observed in AFM FFT images, the enhanced orientational contrast in the xy - and xz -planes at 284.0 eV, and the local maximum ISI at 284.0 eV indicate that the “X”-shaped scattering arises from strain-induced alignment of polymer chains across the fibrillar morphology. *In situ* RSoXS measurements at 285.0 eV during stretching show that “X”-shaped feature disappears (Fig. S44), suggesting reduced orientation sensitivity and further supporting the assignment of scattering contrast origin. Thus, the “X” pattern provides direct information related to the orientation of fibril aggregates.”

Supplementary Information:

Fig. S44. In situ RSoXS images of a P(NDI2OD-T2) thin film at different strains during (a) stretching and (b) releasing, which are probed at 285.0 eV.

To determine the ϕ positions, the azimuthal intensity profiles were fitted using Gaussian functions. For clarity, an example of the peak-fitting is provided in Fig. S47.

Fig. S47. Example of Gaussian-based peak fitting used to extract (a) ϕ_1 and (b) ϕ_2 from azimuthal intensity profiles obtained at 284.0 eV.

(5) *The manuscript suggests that crystallite disruption and increased torsion reduce exciton delocalization, affecting optical absorption. Do the authors have supporting evidence or citations for this claim?*

Our response: Thanks for the comment. To support our interpretation, we performed time-dependent density functional theory (TD-DFT) calculations using dimer structures extracted from the MD simulations of the amorphous polymer under different strain conditions. The results reveal that when the intra-chain dihedral angle increases from 30° to 90° under high strain, the calculated absorption peak exhibits a noticeable blue shift. This trend indicates reduced conjugation and exciton delocalization along the polymer backbone, as well as weakens electronic coupling. To clarify this, we added the following sentence in the manuscript.

Main text (Page 14, line 12):

“Time-dependent density functional theory calculations using strain-dependent dimer structures confirm that increasing the dihedral angle between BT and NDI units from $\sim 30^\circ$ to 90° induces a blue shift in absorption (Fig. S49), consistent with reduced π -conjugation and exciton delocalization.”

Supplementary Information:

Fig. S49. Time-dependent density functional theory (TD-DFT) calculated absorption spectra of strain-dependent dimer structures with different the dihedral angle between BT and NDI units.

(6) Regarding the *J-V* curves of stretchable solar cells, how is the device’s active area defined and measured during strain application? This point should be clarified in the experimental section.

Our response: Thanks for the comment. The initial device area (0.04 cm^2) was defined by the overlap between the patterned M-PH1000 bottom electrode and the patterned EGaIn@Ag top electrode in a cross-electrode geometry. During mechanical stretching, the active area deformed geometrically. To account for this, we corrected the active area by quantifying dimensional changes using optical microscopy. Specifically, the average area change was obtained from three independent devices under identical strain conditions, and the results were fitted with a three-phase exponential decay function (ExpDec3). This fitting curve was then used to correct the active area of the tested devices at each applied strain. To clarify this, we added the following sentences to the

Device Fabrication section.

Method:

“The initial active area (0.04 cm^2) was defined by the overlap of the patterned M-PH1000 bottom electrode and the patterned EGaIn@Ag top electrode in a cross-electrode geometry. During mechanical stretching, the active area deformed geometrically. To account for this, the actual active area was corrected based on dimensional changes measured by optical microscopy. The average area variation was extracted from three independent devices under identical strain conditions and fitted with a three-phase exponential decay function (Fig. S62), which was then applied to correct the active area of the tested devices at each strain.”

Fig. S62. The correction curve for the active area of intrinsically stretchable OPV devices. Black squares denote the averaged active areas of three independent devices, while the red line represents the corresponding fit using a three-phase exponential decay function.

Reviewer #4 (Remarks to the Author):

Reviewer #5 (Remarks to the Author):

Response to Reviewers' Comments

Reviewer #1 (Remarks to the Author):

The authors have adequately addressed the manuscript revisions, and this version has improved significantly. Most of my comments have been addressed with additional experiments and explanations in the text where needed. The only part that remains slightly confusing is the additional AFM analysis for comment 5. The authors introduced power spectral density (PSD) curve analysis of the AFM results, but the results shown in Fig. S41 differ from their description. From the figure, the order of domain size from small to large is 0% (black), 10% (red), 30% (blue), and 20% (yellow), while in the figure caption it was listed as 0%, 10%, 20%, and 30%. Additionally, the authors did not apply the same PSD analysis to their original AFM data from the 50% strain film for comparison. While the manuscript can be published in Nature Communication, I would recommend some additional clarification of the newly added AFM analysis.

Our response: Thanks for the comment. We apologize for the color mismatch in the previously submitted Fig. S41. We have now corrected the dotted-line colors to match the order in the caption: 0% (black), 10% (red), 20% (yellow), and 30% (blue).

To address the second comment, we have also performed the same PSD analysis on the AFM height images of the 50% strained film. The 50% PSD curve shows an overall higher PSD amplitude and a peak shifted to a lower spatial-frequency range, consistent with the formation of larger aggregated domains under high strain. We further refined the calculation of characteristic domain size using the inverse of the spatial frequency k corresponding to the turning or peak positions in the PSD plots. The extracted characteristic domain size increases from 3.6 μm to 6.2 μm , 7.1 μm , 7.7 μm , and 8.3 μm for films strained at 0%, 10%, 20%, 30%, and 50%, respectively. Both the figure and the caption have been updated for clarity as show below.

Fig. S41. Power spectral density (PSD) plots extracted from AFM height images (as shown in Fig. S34 for 0-30% strains and Fig. 3a for 50% strain) of P(NDI2OD-T2) thin films under different stretch strains. The dotted lines highlight the turning or peak positions used to estimate the characteristic domain size, calculated as the inverse of the corresponding spatial frequency k . The characteristic domain size increases from $3.6 \mu\text{m}$ to $6.2 \mu\text{m}$, $7.1 \mu\text{m}$, $7.7 \mu\text{m}$, and $8.3 \mu\text{m}$ for films strained at 0%, 10%, 20%, 30%, and 50%, respectively. These results reflect the micrometer-scale aggregation process captured by AFM.

Reviewer #2 (Remarks to the Author):

The revision has addressed all the comments. I now recommend accepting the paper.

Reviewer #3 (Remarks to the Author):

The authors have thoroughly addressed my comments, and the manuscript has been revised accordingly. Thus, I would recommend it for publication.

Reviewer #4 (Remarks to the Author):

Reviewer #5 (Remarks to the Author):
